# CONSISTENT LOW-RANK APPROXIMATION

**David P. Woodruff**
Carnegie Mellon University
dwoodruf@andrew.cmu.edu

**Samson Zhou**
Texas A&M University
samsonzhou@gmail.com

## ABSTRACT

We introduce and study the problem of consistent low-rank approximation, in which rows of an input matrix $\mathbf{A} \in \mathbb{R}^{n \times d}$ arrive sequentially and the goal is to provide a sequence of subspaces that well-approximate the optimal rank-$k$ approximation to the submatrix $\mathbf{A}^{(t)}$ that has arrived at each time $t$, while minimizing the recourse, i.e., the overall change in the sequence of solutions. We first show that when the goal is to achieve a low-rank cost within an additive $\varepsilon \cdot ||\mathbf{A}^{(t)}||_F^2$ factor of the optimal cost, roughly $\mathcal{O}\left(\frac{k}{\varepsilon} \log(nd)\right)$ recourse is feasible. For the more challenging goal of achieving a relative $(1 + \varepsilon)$-multiplicative approximation of the optimal rank-$k$ cost, we show that a simple upper bound in this setting is $\frac{k^2}{\varepsilon^2} \cdot \operatorname{poly} \log(nd)$ recourse, which we further improve to $\frac{k^{3/2}}{\varepsilon^2} \cdot \operatorname{poly} \log(nd)$ for integer-bounded matrices and $\frac{k}{\varepsilon^2} \cdot \operatorname{poly} \log(nd)$ for data streams with polynomial online condition number. We also show that $\Omega\left(\frac{k}{\varepsilon} \log \frac{n}{k}\right)$ recourse is necessary for any algorithm that maintains a multiplicative $(1 + \varepsilon)$-approximation to the optimal low-rank cost, even if the full input is known in advance. Finally, we perform a number of empirical evaluations to complement our theoretical guarantees, demonstrating the efficacy of our algorithms in practice.

## 1 INTRODUCTION

Low-rank approximation is a fundamental technique that is frequently used in machine learning, data science, and statistics to identify important structural information from large datasets. Given an input matrix $\mathbf{A} \in \mathbb{R}^{n \times d}$ consisting of $n$ observations across $d$ features, the goal of low-rank approximation is to decompose $\mathbf{A}$ into a combination of $k$ independent latent variables called factors. We represent these factors by matrices $\mathbf{U} \in \mathbb{R}^{n \times k}$ and $\mathbf{V} \in \mathbb{R}^{k \times d}$, so that the product $\mathbf{UV}$ should be an "accurate" representation of the original dataset $\mathbf{A}$. Formally, we want to minimize the quantity $\min_{\mathbf{U} \in \mathbb{R}^{n \times k}, \mathbf{V} \in \mathbb{R}^{k \times d}} \mathcal{L}(\mathbf{UV} - \mathbf{A})$, where $\mathcal{L}$ denotes any predetermined loss function chosen for its specific properties. Though there is a variety of standard metrics, such as subspace alignment angles or peak signal-to-noise ratio (PSNR), in this paper we focus on Frobenius loss, which is perhaps the most common loss function, due to its connection with least squares regression. As a result, the problem is equivalent to finding the right factor matrix $\mathbf{V} \in \mathbb{R}^{k \times d}$ of orthonormal rows to minimize the quantity $\|\mathbf{A} - \mathbf{AV}^\top \mathbf{V}\|_F^2$, in which case the optimal left factor matrix is $\mathbf{U} = \mathbf{AV}^\top$.

The rank parameter $k$ is generally chosen based on the complexity of the underlying model chosen to fit the data. The identification and subsequent utilization of the factors can often decrease the number of relevant features in an observation and thus simultaneously improve interpretability and decrease dimensionality. In particular, low-rank approximation facilitates using the factors $\mathbf{U}$ and $\mathbf{V}$ to approximately represent the original dataset $\mathbf{A}$, thus using only $(n + d)k$ parameters in the representation rather than the original $nd$ entries of $\mathbf{A}$. Thereafter, given a vector $\mathbf{x} \in \mathbb{R}^d$, we can compute the matrix-vector product $\mathbf{UVx}$ as an approximation to $\mathbf{Ax}$ in time $(n + d)k$. By contrast, computing $\mathbf{Ax}$ requires $nd$ time. As a result of these advantages and more, low-rank approximation is one of the most common tools, with applications in recommendation systems, mathematical modeling, predictive analytics, dimensionality reduction, computer vision, and natural language processing Chierichetti et al. (2017); Clarkson & Woodruff (2017); Cohen et al. (2017); Musco & Woodruff (2017a;b); Bakshi & Woodruff (2018); Braverman et al. (2021a); Velingker et al. (2023); Musco & Sheth (2024); Song et al. (2024); Kapralov et al. (2026).

**Data streams.** The streaming model of computation has emerged as a popular setting for analyzing large evolving datasets, such as database logs generated from commercial transactions, financial markets, Internet of Things (IoT) devices, scientific observations, social network correspondences, or virtual traffic measurements. In the row-arrival model, each stream update provides an additional observation, i.e., an additional matrix row $\mathbf{a}_t \in \mathbb{R}^d$ of the ultimate input $\mathbf{A} \in \mathbb{R}^{n \times d}$ for low-rank approximation. Often, downstream decisions and policies must be determined with uncertainty about future inputs. That is, the input may be revealed in a data stream and practitioners may be required to make irrevocable choices at time $t \in [n]$, given only the dataset $\mathbf{A}^{(t)} = \mathbf{a}_1 \circ \ldots \circ \mathbf{a}_t$ consisting of the first $t$ data point observations. A prototypical example is the setting of online caching/paging algorithms (Fiat et al., 1991; Irani, 1996), which must choose to keep or evict items in the cache at every time step. This generalizes to the $k$-server problem for penalties that are captured by metric spaces (Manasse et al., 1990; Bansal et al., 2015; Bubeck et al., 2023).

**Consistency.** Although irrevocable decisions are theoretically interesting in the context of online algorithms, such a restriction may be overly stringent for many practical settings. Lattanzi & Vassilvitskii (2017) notes that in the earlier setting of caching and paging, a load balancer that receives online requests for assignment to different machines can simply reassign some of the past tasks to other machines to increase overall performance if necessary. Moreover, the ability of algorithms to adjust previous decisions based on updated information allows for a richer understanding of the structure of the underlying problem beyond the impossibility barriers of online algorithms. On the other hand, such adjustments have downstream ramifications to applications that rely on the decisions/policies resulting from the algorithm. Therefore, the related notions of consistency and recourse were defined to quantify the cumulative number of changes to the output solution of an algorithm over time. Low-recourse online algorithms have been well-studied for a number of problems (Gupta & Kumar, 2014; Gupta et al., 2014; Lacki et al., 2015; Gu et al., 2016; Megow et al., 2016; Gupta & Levin, 2020; Bhattacharya et al., 2023), while consistent clustering and facility location have recently received considerable attention (Lattanzi & Vassilvitskii, 2017; Cohen-Addad et al., 2019; Fichtenberger et al., 2021; Lacki ⓡ et al., 2023). In a standard scenario in feature engineering, low-rank approximation algorithms are used to select specific features or linear combinations of features, on which models are trained. Thus, large consistency costs correspond to expensive retrainings of whole large-scale machine learning systems (Lattanzi & Vassilvitskii, 2017), due to different sets of features being passed to the learner.

Building on the notion of consistency, we formalize the problem of consistent low-rank approximation, which captures the need for online algorithms that adapt to evolving data while keeping their outputs stable for downstream use. As discussed above, low-rank approximation is widely used for feature engineering: the factors produced define the feature subspace on which downstream models are trained. In dynamic settings, high recourse can cause these features to change abruptly even under minor updates, triggering frequent and costly retraining of large-scale models (Lattanzi & Vassilvitskii, 2017). Consistent, low-recourse LRA algorithms maintain feature stability over time, reducing retraining costs and improving reliability.

The importance of stability extends across many practical domains. In biometrics, consistent low-dimensional representations of fingerprints or iris patterns are crucial for maintaining reliable identification (Jain et al., 2004). In image processing, tasks such as object detection, handwriting recognition, and facial recognition depend on derived features that should not fluctuate unpredictably as new data arrives (Nixon & Aguado, 2012). In data compression and signal processing, stable reduced representations preserve essential structure while controlling noise (Witten & Frank, 2002; Proakis, 2007). In text mining and information retrieval, numerical features such as TF–IDF vectors or embeddings must remain coherent to maintain the quality of search and classification (Aggarwal & Aggarwal, 2015; Schütze et al., 2008). Even in large-scale data curation for foundation models—where clustering, a constrained form of low-rank approximation, is used to deduplicate data—high recourse leads to unstable representative sets and repeated retraining of models (Lacki ⓡ et al., 2023).

Across these settings, the underlying principle is consistent: downstream systems rely not only on the quality of the approximation but also on the *stability* of the feature representations over time. By explicitly accounting for this need, consistent low-rank approximation provides solutions that evolve smoothly while maintaining strong accuracy guarantees, delivering both theoretical insight and concrete practical benefits in dynamic, real-world pipelines.

## 1.1 Our Contributions

In this paper, we initiate the study of consistent low-rank approximation.

**Formal model.** Given an accuracy parameter $\varepsilon > 0$, our goal is to provide a $(1 + \varepsilon)$-approximation to low-rank approximation at all times. We assume the input is a matrix $\mathbf{A} \in \mathbb{R}^{n \times d}$, whose rows $\mathbf{a}_1, \ldots, \mathbf{a}_n$ arrive sequentially, so that at each time $t$, the algorithm only has access to $\mathbf{A}^{(t)}$, the first $t$ rows of $\mathbf{A}$. That is, the goal of the algorithm is firstly to output a set $\mathbf{V}^{(t)} \in \mathbb{R}^{k \times d}$ of $k$ factors at each time $t \in [n]$, so that

$$\|\mathbf{A}^{(t)} - \mathbf{A}^{(t)}(\mathbf{V}^{(t)})^\top \mathbf{V}^{(t)}\|_F^2 \leq (1 + \varepsilon) \cdot \mathsf{OPT}_t,$$

where $\mathsf{OPT}_t$ is the cost of the optimal low-rank approximation at time $t$, $\mathsf{OPT}_t = \min_{\mathbf{V} \in \mathbb{R}^{k \times d}} \|\mathbf{A}^{(t)} - \mathbf{A}^{(t)}\mathbf{V}^\top \mathbf{V}\|_F^2$. In other words, we want the low-rank cost induced by the factor $\mathbf{V}^{(t)}$ returned by the algorithm to closely capture the optimal low-rank approximation. Secondly, we would like the sequence $\mathbf{V}^{(1)}, \ldots, \mathbf{V}^{(n)}$ of factors to change minimally over time. Specifically, the goal of the algorithm is to minimize $\sum_{t=2}^n \mathrm{Recourse}(\mathbf{V}^{(t)}, \mathbf{V}^{(t-1)})$, where $\mathrm{Recourse}(\mathbf{R}, \mathbf{T}) = \|\mathbf{P_R} - \mathbf{P_T}\|_F^2$ is the squared subspace distance between the orthogonal projection matrices $\mathbf{P_R}, \mathbf{P_T}$ of the two subspaces.

We remark on our choice of the cost function $\mathrm{Recourse}(\mathbf{R}, \mathbf{T})$ for factors $\mathbf{R}$ and $\mathbf{T}$. At first glance, a natural setting of the cost function may be the number of vectors that are different between $\mathbf{R}$ and $\mathbf{T}$, since in some sense it captures the change between $\mathbf{R}$ and $\mathbf{T}$. However, it should be observed that even if there is a unique rank $k$ subspace $\mathbf{V}$ that minimizes the low-rank approximation cost $\|\mathbf{A} - \mathbf{A}\mathbf{V}^\top \mathbf{V}\|_F^2$, there may be many representations of $\mathbf{V}$, up to any arbitrary rotation of the basis vectors within the subspace. Thus, a cost function sensitive to the choice of basis vectors may not be appropriate because a large change in the change of basis vectors may not result in any change in the resulting projection $\mathbf{A}\mathbf{V}^\top \mathbf{V}$. This implies that a reasonable cost function should capture the difference in the spaces spanned by the subspaces $\mathbf{R}$ and $\mathbf{T}$. For example, the dimension of the subspace of $\mathbf{T}$ that is orthogonal to $\mathbf{R}$ would be an appropriate quantity. However, it should be noted that this quantity still punishes a subspace $\mathbf{T}$ that is a small perturbation of $\mathbf{R}$, for example if $\mathbf{R}$ is the elementary vector $(0, 1)$ and $\mathbf{T}$ is the vector $(\varepsilon, \sqrt{1 - \varepsilon^2})$ for arbitrarily small $\varepsilon$. A more robust quantity would be a continuous analogue of the dimension, which is the squared mass of the projection of $\mathbf{T}$ away from $\mathbf{R}$; this quantity corresponds exactly to our cost function $\mathrm{Recourse}$. Thus, we believe that our choice of the consistency cost function is quite natural.

We note that we can further assume that the input matrix $\mathbf{A}$ has integer entries bounded in magnitude by some parameter $M$. We remark that this assumption is standard in numerical linear algebra because in general it is difficult to represent real numbers up to arbitrary precision in the input of the algorithm. Instead, for inputs that are rational, after appropriate scaling each entry of the input matrix can be written as an integer. Thus, this standard assumption can model the number of bits used to encode each entry of the matrix, without loss of generality.

**Theoretical results.** We first note that the optimal low-rank approximation can completely change at every step, in the sense that the optimal subspace $\mathbf{V}^{(t-1)}$ at time $t - 1$ may still have dimension $k$ after being projected onto the optimal subspace $\mathbf{V}^{(t)}$. Thus, to achieve optimality, it may be possible that $\Omega(nk)$ recourse could be necessary, i.e., by recomputing the best $k$ factors after the arrival of each of the $n$ rows. Nevertheless, on the positive side, we first show sublinear recourse can be achieved if the goal is to simply achieve an additive $\varepsilon \cdot \|\mathbf{A}^{(t)}\|_F^2$ additive error to the low-rank approximation cost at all times.

**Theorem 1.1.** *Suppose $\mathbf{A} \in \mathbb{Z}^{n \times d}$ is an integer matrix with rank $r > k$ and entries bounded in magnitude by $M$ and let $\mathbf{A}^{(t)}$ denote the first $t$ rows of $\mathbf{A}$, for any $t \in [n]$. There exists an algorithm that achieves $\varepsilon \cdot \|\mathbf{A}^{(t)}\|_F^2$-additive approximation to the cost of the optimal low-rank approximation $\mathbf{A}$ at all times and achieves recourse $\mathcal{O}\left(\frac{k}{\varepsilon} \log(ndM)\right)$.*

We remark that the algorithm corresponding to Theorem 1.1 uses $\frac{kd}{\varepsilon} \cdot \mathrm{polylog}(ndM)$ bits of space and $d \cdot \mathrm{poly}\left(k, \frac{1}{\varepsilon}, \log(ndM)\right)$ amortized update time. Since the squared Frobenius norm is an upper bound on the optimal low-rank approximation cost, achieving additive $\varepsilon \cdot \|\mathbf{A}^{(t)}\|_F^2$ error to the optimal cost is significantly easier than achieving relative $(1 + \varepsilon)$-multiplicative error, particularly

in the case where the top singular vectors correspond to large singular values. In fact, we can even achieve recourse linear in $k$ if the online condition number of the stream is at most $\text{poly}(n)$:

**Theorem 1.2.** *Given a stream with online condition number* $\text{poly}(n)$*, there exists an algorithm that achieves a* $(1 + \varepsilon)$*-approximation for low-rank approximation, and uses recourse* $\mathcal{O}\left(\frac{k}{\varepsilon^2} \log^3 n\right)$.

We remark that for streams with online condition number $\text{poly}(n)$, we actually show a stronger result in Lemma 2.1. In particular, we show that the optimal rank-$k$ subspace changes by only a constant amount after rank-one perturbations corresponding to single-entry changes, row modifications, row insertions, and row deletions. Thus this result implies Theorem 1.2 using standard techniques for reducing the "effective" stream length and in fact, also immediately gives an algorithm that maintains the optimal rank-$k$ approximation under any sequence of such updates while incurring only $\mathcal{O}(n)$ total recourse over a stream of $n$ operations. Hence, our approach can handle explicit distributional shifts arising in insertion–deletion streams; extending this ability to handle implicit deletions such as in the sliding window model (Datar et al., 2002; Crouch et al., 2013; Braverman et al., 2021b; 2018; 2020; Woodruff & Zhou, 2021; Borassi et al., 2020; Epasto et al., 2022; Jayaram et al., 2022; Blocki et al., 2023; Woodruff & Yasuda, 2023; Woodruff et al., 2023; Cohen-Addad et al., 2025; Braverman et al., 2026; Nagawanshi et al., 2026) is a natural direction for future work.

For standard matrices with integer entries bounded by $\text{poly}(n)$, however, the assumption that the online condition number is bounded by $\text{poly}(n)$ may not hold. For example, it is known that there exist integer matrices with dimension $n \times d$ but optimal low-rank cost as small as $\exp(-\Omega(k))$. To that end, we first observe that a simple application of a result by Braverman et al. (2020) can be used to achieve recourse $\frac{k^2}{\varepsilon^2} \cdot \text{polylog}(ndM)$ while maintaining a $(1 + \varepsilon)$-multiplicative approximation at all times. Indeed, for constant $\varepsilon$, roughly $k \cdot \text{polylog}(ndM)$ rows can be sampled through a process known as online ridge-leverage score sampling, to preserve the low-rank approximation at all times. Then for quadratic recourse, it suffices to recompute the top right $k$ singular vectors for the sampled submatrix each time a new row is sampled. A natural question is whether $\Omega(k)$ recourse is necessary for each step, i.e., whether recomputing the top right $k$ singular vectors is necessary. We show this is not the case, and that overall the recourse can be made sub-quadratic.

**Theorem 1.3.** *Suppose* $\mathbf{A} \in \mathbb{Z}^{n \times d}$ *is an integer matrix with entries bounded in magnitude by $M$. There exists an algorithm that achieves a* $(1 + \varepsilon)$*-approximation to the cost of the optimal low-rank approximation* $\mathbf{A}$ *at all times and achieves recourse* $\frac{k^{3/2}}{\varepsilon^2} \cdot \text{polylog}(ndM)$.

We again remark that the algorithm corresponding to Theorem 1.3 uses $\frac{kd}{\varepsilon} \cdot \text{polylog}(ndM)$ bits of space and $d \cdot \text{poly}\left(k, \frac{1}{\varepsilon}, \log(ndM)\right)$ amortized update time. Finally, we show that $\Omega\left(\frac{k}{\varepsilon} \log \frac{n}{k}\right)$ recourse is necessary for any multiplicative $(1 + \varepsilon)$-approximation algorithm for low-rank approximation, even if the full input is known in advance.

**Theorem 1.4.** *For any parameter* $\varepsilon > \frac{\log n}{n}$*, there exists a sequence of rows* $\mathbf{x}_1, \ldots, \mathbf{x}_n \in \mathbb{R}^d$ *such that any algorithm that produces a* $(1 + \varepsilon)$*-approximation to the cost of the optimal low-rank approximation at all times must have consistency cost* $\Omega\left(\frac{k}{\varepsilon} \log \frac{n}{k}\right)$.

**Empirical evaluations.** We complement our theoretical results with a number of empirical evaluations in Section 4. Our results show that although our formal guarantees provide a worst-case analysis of the approximation cost of the low-rank solution output by our algorithm, the performance can be even (much) better in practice. Importantly, our results show that algorithms for online low-rank approximation such as Frequent Directions (Ghashami et al., 2016) do not achieve good recourse, motivating the study of algorithms specifically designed for consistent low-rank approximation.

**Organization of the paper.** We give the linear recourse algorithms in Section 2 and conduct empirical evaluations in Section 4 and Appendix G. We give our result for integer-valued matrices in Section 3. We defer all proofs to the full appendix, and specifically the lower bound to Appendix D. The reader may also find it helpful to consult Appendix A for standard notation and additional preliminaries used in our paper.

## 1.2 RELATED WORK

In this section, we briefly describe a number of existing techniques in closely related models and provide intuition on why they do not suffice for our setting.

**Frequent directions and online ridge leverage score sampling.** The most natural approach would be to apply existing algorithms from the streaming literature for low-rank approximation. The two most related works are the deterministic Frequent Directions work by Ghashami et al. (2016) and the (online) ridge leverage score sampling procedure popularized by Cohen et al. (2017); Braverman et al. (2020). Both procedures maintain a small number of rows that capture the "important" directions of the matrix at all times. Hence to report a near-optimal rank-$k$ approximation at each time, these algorithms simply return the top $k$ right singular vectors of the singular value decomposition of the matrix stored by each algorithm. However, one could easily envision a situation in which the $k$-th and $(k+1)$-th largest singular vectors repeatedly alternate, incurring recourse at each step. For example, suppose $k = 1$ and at all times $2t$ for integral $t > 0$, the top singular vectors are $(2t, 0)$ and $(0, 2t - \varepsilon)$, but at all times $2t + 1$ for integral $t \geq 0$, the top singular vectors are $(2t + 1 - \varepsilon, 0)$ and $(0, 2t + 1)$. Then at all times $2t$, the best rank-$k$ solution for $k = 1$ would be the elementary vector $\mathbf{e}_1$ while at all times $2t + 1$, the best rank-$k$ solution would be the elementary vector $\mathbf{e}_2$. These algorithms would incur recourse $n$, whereas even an algorithm that never changes the initial vector $\mathbf{e}_2$ would incur recourse $0$. Thus, these algorithms seem to fail catastrophically, i.e., not even provide a $\mathrm{poly}(n)$-multiplicative approximation to the recourse, even for simple inputs.

One may observe that our goal is to only upper bound the total recourse, rather than to achieve a multiplicative approximation to the recourse. Indeed for this purpose, the online ridge leverage sampling technique provides some gain. In particular, Braverman et al. (2020) showed that over the entirety of the stream, at most $\frac{k}{\varepsilon^2} \cdot \mathrm{polylog}\left(n, d, \frac{1}{\varepsilon}\right)$ rows will be sampled into the sketch matrix in total, and moreover the sketch matrix will accurately capture the residual for the projection onto any subspace of dimension $k$. Thus to achieve $\mathcal{O}\left(k^2\right)$ recourse, it suffices to simply recompute the top-$k$ singular vectors each time a new row is sampled by the online ridge leverage sampling procedure.

**Singular value decomposition.** Note that the previous example also shows that more generally, it does not suffice to simply output the top-$k$ right singular vectors of the singular value decomposition (SVD). However, one might hope that it suffices to replace just a single direction in the SVD each time a row is sampled into the sketch matrix by online ridge leverage score sampling. Unfortunately, it seems possible that an approximately optimal solution from a previous step could require all $k$ factors to be replaced by the arrival of a single row. Suppose for example, that the factor $\mathbf{V}^{(1)}$ consisting of the elementary vectors $\mathbf{e}_{k+1}, \ldots, \mathbf{e}_{2k}$ achieves the same loss as the factor $\mathbf{V}^{(2)}$ consisting of the elementary vectors $\mathbf{e}_1, \ldots, \mathbf{e}_k$. Now if the next row is non-zero exactly in the coordinates $1, \ldots, k$, then the top $k$ space could change entirely, from $\mathbf{V}^{(1)}$ to $\mathbf{V}^{(2)}$. While this worst-case input is unavoidable, we show this can only happen a small number of times. It is more problematic when only one of these factors drastically change, while the other $k - 1$ factors only change by a little, but we still output $k$ completely new factors. Our algorithm avoids this by carefully choosing the factors to replace based on casework on the corresponding singular values.

## 2 SIMPLE ALGORITHMS WITH OPTIMAL RECOURSE

In this section, we briefly describe two simple algorithms that achieve recourse linear in $k$.

**Additive error.** The first algorithm roughly $\mathcal{O}\left(\frac{k}{\varepsilon} \log n\right)$ recourse when the goal is to maintain an additive $\varepsilon \cdot \|\mathbf{A}^{(t)}\|_F^2$ error at all times $t \in [n]$, where $\mathbf{A}^{(t)}$ is the $t$ rows of matrix $\mathbf{A}$ that have arrived at time $t$. We simply track the squared Frobenius norm of the matrix $\mathbf{A}^{(t)}$ at all times. For each time the squared Frobenius norm has increased by a factor of $(1 + \varepsilon)$, then we recompute the singular value decomposition of $\mathbf{A}^{(t)}$ and choose $\mathbf{V}^{(t)}$ to be the top $k$ right singular vectors of $\mathbf{A}^{(t)}$. For all other times, we maintain the same set of factors.

The main intuition is that each time we reset $\mathbf{V}^{(t)}$, we find the optimal solution at time $t$. Over the next few steps after $t$, our solution will degrade, but the most it can degrade by is the squared Frobenius norm of the submatrix formed by the incoming rows. Thus as long as this quantity is less than an $\varepsilon$-fraction of the squared Frobenius norm of the entire matrix, then our correctness guarantee will hold. On the other hand, such a guarantee *must* hold as long as the squared Frobenius norm has not increased by a $(1 + \varepsilon)$-multiplicative factor. Thus we incur recourse $k$ for each of the $\mathcal{O}\left(\frac{1}{\varepsilon} \log(ndM)\right)$ times the matrix can have its squared Frobenius norm increase by $(1 + \varepsilon)$ multiplicatively. We give our algorithm in full in Algorithm 4 in Appendix E.

**Bounded online condition number.** We next show that the optimal rank $k$ subspace incurs at most constant recourse under rank one perturbations. In particular, it suffices to consider the case where a single row is added to the matrix:

**Lemma 2.1.** *Let $\mathbf{A}^{(t-1)} \in \mathbb{R}^{(t-1) \times d}$ and $\mathbf{A}^{(t)} \in \mathbb{R}^{t \times d}$ such that $\mathbf{A}^{(t)}$ is $\mathbf{A}^{(t-1)}$ with the row $\mathbf{A}_t$ appended. Let $\mathbf{V}_{t-1}^*$ and $\mathbf{V}_t^*$ be the optimal rank-$k$ subspaces (the span of the top $k$ right singular vectors) of $\mathbf{A}^{(t-1)}$ and $\mathbf{A}^{(t)}$, respectively. Then $\mathrm{Recourse}(\mathbf{V}_{t-1}^*, \mathbf{V}_t^*) \leq 8$.*

Note that whenever a single entry of a matrix is changed, any row of a matrix is changed, a new row of a matrix is added, or an existing row of a matrix is deleted, these all correspond to rank one perturbations of the matrix. Thus, we immediately have the following corollary:

**Theorem 2.2.** *There exists an algorithm that maintains the optimal rank-$k$ approximation of a matrix under any sequence of rank-one updates, including entry modifications, row modifications, row insertions, or row deletions, and incurs total recourse $\mathcal{O}(n)$ on a stream of $n$ updates. Moreover, if each row has dimension $d$, then the update time is $\mathcal{O}\left((n+d)k + k^3\right)$.*

Given the statement in Lemma 2.1, we immediately obtain an algorithm with $\mathcal{O}(n)$ recourse in the row-arrival model for a stream of length $n$. Because $n$ recourse is too large, we use the following standard approach to decrease the effective number of rows in the matrix.

**Definition 2.3** (Projection-cost preserving sketch). *Given a matrix $\mathbf{A} \in \mathbb{R}^{n \times d}$, a matrix $\mathbf{M} \in \mathbb{R}^{m \times d}$ is a $(1 + \varepsilon)$ projection-cost preserving sketch of $\mathbf{A}$ if for all projection matrices $\mathbf{P} \in \mathbb{R}^{d \times d}$,*

$$(1 - \varepsilon) \|\mathbf{A} - \mathbf{AP}\|_F^2 \leq \|\mathbf{M} - \mathbf{MP}\|_F^2 \leq (1 + \varepsilon) \|\mathbf{A} - \mathbf{AP}\|_F^2.$$

Intuitively, a projection-cost preserving sketch is a sketch matrix that approximately captures the residual mass after projecting away any rank $k$ subspace. The following theorem shows that a projection-cost preserving sketch can be acquired via online ridge leverage sampling.

**Theorem 2.4** (Theorem 3.1 in Braverman et al. (2020)). *Given an accuracy parameter $\varepsilon > 0$, a rank parameter $k > 0$, and a matrix $\mathbf{A} = \mathbf{a}_1 \circ \ldots \circ \mathbf{a}_n \in \mathbb{R}^{n \times d}$ whose rows arrive sequentially in a stream with condition number $\kappa$, there exists an algorithm that outputs a matrix $\mathbf{M}$ with $\mathcal{O}\left(\frac{k}{\varepsilon^2} \log n \log^2 \kappa\right)$ rescaled rows of $\mathbf{A}$ such that*

$$(1 - \varepsilon) \|\mathbf{A} - \mathbf{A}_{(k)}\|_F^2 \leq \|\mathbf{M} - \mathbf{M}_{(k)}\|_F^2 \leq (1 + \varepsilon) \|\mathbf{A} - \mathbf{A}_{(k)}\|_F^2,$$

*so that with high probability, $\mathbf{M}$ is a rank $k$ projection-cost preservation of $\mathbf{A}$.*

In particular, if the online condition number of the stream is upper bounded by $\mathrm{poly}(n)$, then Theorem 2.4 states that online ridge leverage sampling achieves an online coreset of size $\mathcal{O}\left(\frac{k}{\varepsilon^2} \log^3 n\right)$. By applying Lemma 2.1, it follows that simply maintaining the optimal rank-$k$ subspace for the online coreset at all times, we have Theorem 1.2.

## 3 ALGORITHM FOR RELATIVE ERROR

In this section, we give our algorithm for $(1 + \varepsilon)$-multiplicative relative error at all times in the stream. Let $t \in [n]$, let $\mathbf{x}_t \in \{-\Delta, \ldots, \Delta - 1, \Delta\}^d$ and let $\mathbf{X}_t = \mathbf{x}_1 \circ \ldots \circ \mathbf{x}_t$. For each $\mathbf{X}_t \in \mathbb{R}^{t \times d}$, we compute a low-rank approximation $\mathbf{U}_t \mathbf{V}_t$ of $\mathbf{X}_t$, where $\mathbf{U}_t \in \mathbb{R}^{t \times k}$ and $\mathbf{V}_t \in \mathbb{R}^{k \times d}$ are the factors of $\mathbf{X}_t$. We abuse notation and write $\mathbf{V}_t$ as a set $V_t$ of $k$ points in $\mathbb{R}^d$. Note that the quantity $\sum_{t=1}^n |V_t \setminus V_{t-1}|$ is an upper bound on the recourse or the consistency cost. Thus in this section, we interchangeably refer to this quantity as the recourse or the consistency cost, as we can lower bound this sharper quantity.

First, we recall an important property about the optimal solution for our formulation of low-rank approximation, i.e., with Frobenius loss.

**Theorem 3.1** (Eckart-Young-Mirsky theorem). *(Eckart & Young, 1936; Mirsky, 1960) Let $\mathbf{A} \in \mathbb{R}^{n \times d}$ with rank $r$ have singular value decomposition $\mathbf{A} = \mathbf{U}\boldsymbol{\Sigma}\mathbf{V}$ for $\mathbf{U} \in \mathbb{R}^{n \times r}$, $\boldsymbol{\Sigma} \in \mathbb{R}^{r \times r}$, $\mathbf{V} \in \mathbb{R}^{r \times d}$ and singular values $\sigma_1(\mathbf{A}) \geq \sigma_2(\mathbf{A}) \geq \ldots \geq \sigma_d(\mathbf{A})$. Let $\mathbf{X}$ be the top $k$ right singular vectors of $\mathbf{A}$, i.e., the top $k$ rows of $\mathbf{V}$, breaking ties arbitrarily. Then an optimal rank $k$ approximation of $\mathbf{A}$ is $\mathbf{AX}^\top \mathbf{X}$ and the cost is $\|\mathbf{A} - \mathbf{AX}^\top \mathbf{X}\|_F^2 = \sum_{i=k+1}^d \sigma_i^2(\mathbf{A})$.*

As a corollary to Theorem 3.1, we have that an algorithmic procedure to compute an optimal rank $k$ approximation is just to take the top $k$ right singular vectors of $\mathbf{A}$, c.f., procedure RECLUSTER$(\mathbf{A}, k)$ in Algorithm 1, though faster methods for approximate SVD can also be used.

**Corollary 3.2.** *There exists an algorithm* RECLUSTER$(\mathbf{A}, k)$ *that outputs a set of orthonormal rows* $\mathbf{X}$ *that produces the optimal rank $k$ approximation to* $\mathbf{A}$.

---

**Algorithm 1** RECLUSTER$(\mathbf{A}, k)$, i.e., Truncated SVD

---

**Input:** Matrix $\mathbf{A} \in \mathbb{R}^{n \times d}$, rank parameter $k$
**Output:** Top $k$ right singular vectors of $\mathbf{A}$
1: Let $r$ be the rank of $\mathbf{A}$
2: Let $\mathbf{U} \in \mathbb{R}^{n \times r}, \boldsymbol{\Sigma} \in \mathbb{R}^{r \times r}, \mathbf{V} \in \mathbb{R}^{r \times d}$ be the singular value decomposition of $\mathbf{A} = \mathbf{U} \boldsymbol{\Sigma} \mathbf{V}$
3: Return the first $\min(r, k)$ rows of $\mathbf{V}$

---

Our algorithm performs casework on the contribution of the bottom $\sqrt{k}$ singular values of the top $k$. If the contribution is small, the corresponding singular vectors can be replaced without substantially increasing the error. Thus, each time a new row arrives, we simply replace one of the bottom singular vectors with the new row. On the other hand, if the contribution is large, it means that the optimal solution cannot be projected away too much from these directions, or else the optimal low-rank approximation cost will also significantly increase. Thus we can simply choose the optimal set of top right $k$ singular vectors at each time, because there will be substantial overlap between the new subspace and the old subspace. We give our algorithm in full in Algorithm 2.

---

**Algorithm 2** Relative-error algorithm for low-rank approximation with low recourse

---

**Input:** Rows $\mathbf{a}_1, \ldots, \mathbf{a}_n$ of input matrix $\mathbf{A} \in \mathbb{R}^{n \times d}$ with integer entries bounded in magnitude by $M$
**Output:** $\left(1 + \frac{\varepsilon}{4}\right)$-approximation to the cost of the optimal low-rank approximation at all times
1: **for** each row $\mathbf{a}_t$ **do**
2:   $\mathsf{OPT} \leftarrow \sum_{i=k+1}^{d} \sigma_i^2(\mathbf{A}_t)$
3:   $\mathsf{HEAVY} \leftarrow \mathsf{TRUE}, C \leftarrow 0, c \leftarrow 0$
4:   **if** $\mathsf{OPT} \geq \left(1 + \frac{\varepsilon}{4}\right) C$ or ($c = \sqrt{k}$ and $\mathsf{HEAVY} = \mathsf{FALSE}$) or ($c = k$ and $\mathsf{HEAVY} = \mathsf{TRUE}$) **then**
5:     $C \leftarrow \mathsf{OPT}, c \leftarrow 0$
6:     $\mathbf{V} \leftarrow$ RECLUSTER$(\mathbf{A}^{(t)}, k)$
7:     **if** $\sum_{i=k-\sqrt{k}}^{k} \sigma_i^2(\mathbf{A}^{(t)}) \geq \frac{\varepsilon}{3} \cdot C$ **then**
8:       $\mathsf{HEAVY} \leftarrow \mathsf{TRUE}$
9:     **else**
10:       $\mathsf{HEAVY} \leftarrow \mathsf{FALSE}$
11:     **end if**
12:   **else**
13:     **if** $\mathsf{HEAVY} = \mathsf{FALSE}$ **then**
14:       Let $\mathbf{v}$ be the unit vector in $\mathbf{V}$ that minimizes $\|\mathbf{A}^{(s)} \mathbf{v}\|_2^2$, where $s$ was the most recent time RECLUSTER was called
15:       Replace $\mathbf{v}$ with $\frac{1}{\|\mathbf{a}_t\|_2} \cdot \mathbf{a}_t$ in $\mathbf{V}$                    ▷Ignore all zero rows
16:       $c \leftarrow c + 1$
17:     **else if** $\mathsf{HEAVY} = \mathsf{TRUE}$ **then**
18:       **if** $\|\mathbf{A}^{(t)} - \mathbf{A}^{(t)} \mathbf{V}^\top \mathbf{V}\|_F^2 \geq \left(1 + \frac{\varepsilon}{2}\right) \mathsf{OPT}$ **then**
19:         $\mathbf{V} \leftarrow$ RECLUSTER$(\mathbf{A}^{(t)}, k)$
20:         $c \leftarrow c + 1$
21:       **end if**
22:     **end if**
23:   **end if**
24:   Return $\mathbf{V}$
25: **end for**

---

**Correctness.**    For the purposes of discussion, say that an epoch is the set of times during which the optimal low-rank approximation cost has not increased by a multiplicative $(1 + \varepsilon)$-approximation. We first show the correctness of our algorithm across the times $t$ during epochs in which HEAVY is set to FALSE. That is, we show that our algorithm maintains a $(1 + \varepsilon)$-multiplicative approximation to the optimal low-rank approximation cost across all times $t$ in an epoch where the bottom $\sqrt{k}$ singular values of the top $k$ right singular values do not contribute significant mass.

**Lemma 3.3.** *Consider a time $s$ during which $c$ is reset to $0$. Suppose* HEAVY *is set to* FALSE *at time $s$ and $c$ is not reset to $0$ within the next $r$ steps, for $r \le \sqrt{k}$. Let $\mathbf{V}^{(t)}$ be the output of $\mathbf{V}$ at time $t$. Then $\mathbf{V}^{(t)}$ provides a $\left(1 + \frac{\varepsilon}{2}\right)$-approximation to the cost of the optimal low-rank approximation of $\mathbf{A}^{(t)}$ for all $t \in [s, s + r]$.*

Next, we show the correctness of our algorithm across the times $t$ during epochs in which HEAVY is set to TRUE. That is, we show that our algorithm maintains a $(1 + \varepsilon)$-multiplicative approximation to the optimal low-rank approximation cost across all times $t$ in an epoch where the bottom $\sqrt{k}$ singular values of the top $k$ right singular values do contribute significant mass.

**Lemma 3.4.** *Consider a time $t$ during which* HEAVY *is set to* TRUE. *Let $\mathbf{V}^{(t)}$ be the output of $\mathbf{V}$ at time $t$. Then $\mathbf{V}^{(t)}$ provides a $\left(1 + \frac{\varepsilon}{2}\right)$-approximation to the cost of the optimal low-rank approximation of $\mathbf{A}^{(t)}$.*

Correctness at all times now follows from Lemma 3.3 and Lemma 3.4:

**Lemma 3.5.** *At all times $t \in [n]$, Algorithm 2 provides a $\left(1 + \frac{\varepsilon}{2}\right)$-approximation to the cost of the optimal low-rank approximation of $\mathbf{A}^{(t)}$.*

**Recourse.**    We first bound the recourse if the bottom $\sqrt{k}$ singular values of the top $k$ right singular values do not contribute significant mass.

**Lemma 3.6.** *Suppose* HEAVY *is set to* FALSE *at time $s$ and $c$ is reset to $0$ at time $s$. If $c$ is not reset to $0$ within the next $r$ steps, for $r \le \sqrt{k}$, then $\sum_{i=s+1}^{s+r} \text{Recourse}(\mathbf{V}^{(s)}, \mathbf{V}^{(s-1)}) \le r$.*

At this point, we remark a subtlety in the analysis that is easily overlooked. Our general strategy is to show that each time the cost of the optimal low-rank approximation doubles, we should incur recourse $\mathcal{O}\left(k^{1.5}\right)$. One might then expect that because the matrix contains integer entries bounded by $\text{poly}(n)$, then the cost of the optimal low-rank approximation can only double $\mathcal{O}\left(\log n\right)$ times, since it can be at most $\text{poly}(n)$.

Unfortunately, there exist constructions of anti-Hadamard integer matrices with dimension $n \times d$ but optimal low-rank cost as small as $\exp(-\Omega(k))$. Hence, the optimal cost can double $\mathcal{O}\left(k\right)$ times, thereby incurring total recourse $\mathcal{O}\left(k^{2.5}\right)$, which is undesirably large. Instead, we show that when the optimal low-rank cost is exponentially small, then the rank of the matrix must also be quite small, meaning that the recourse of our algorithm cannot be as large as in the full-rank case. To that end, we require a structural property, c.f., Lemma F.2 that describes the cost of the optimal low-rank approximation, and parameterized to handle general matrices with rank $r > k$. This is to handle the case where the cost of the optimal low-rank approximation may be exponentially small in $k$. As a result, we have the following upper bound on the total recourse across all epochs when the bottom $\sqrt{k}$ singular values of the top $k$ do not contribute significant mass.

**Lemma 3.7.** *Suppose* HEAVY *is set to* TRUE *at time $t$ and $c$ is not reset to $0$ within the next $r$ steps, for $r \le k$. Then $\sum_{i=t+1}^{t+r} \text{Recourse}(\mathbf{V}^{(t)}, \mathbf{V}^{(t-1)}) \le r\sqrt{k}$.*

We analyze the total recourse during times when we reset the counter $c$ because the cost of the optimal low-rank approximation has doubled. Finally, it remains to bound the total recourse at times when we transition from one epoch to another. Specifically, we bound the recourse at times $t$ where the optimal low-rank approximation cost has increased by a multiplicative $(1 + \varepsilon)$-approximation since the beginning of the previous epoch.

**Lemma 3.8.** *Let $T$ be the set of times at which $c$ is set to $0$. Then $\sum_{t \in T} \text{Recourse}(\mathbf{V}^{(t)}, \mathbf{V}^{(t-1)}) \le \mathcal{O}\left(n\sqrt{k} + \frac{k}{\varepsilon}\log^2(ndM)\right)$.*

Using Lemma 3.6, Lemma 3.7, and Lemma 3.8, we then bound the total recourse of our algorithm.

**Lemma 3.9.** *The total recourse of Algorithm 2 on an input matrix* $\mathbf{A} \in \mathbb{R}^{n \times d}$ *with integer entries bounded in magnitude by* $M$ *is* $\mathcal{O}\left(n\sqrt{k} + \frac{k}{\varepsilon}\log^2(ndM)\right)$.

Using Lemma 3.5 and Lemma 3.9, we can provide the formal guarantees of our subroutine.

**Lemma 3.10.** *Given an input matrix* $\mathbf{A} \in \mathbb{R}^{n \times d}$ *with integer entries bounded in magnitude by* $M$, *Algorithm 2 achieves a* $\left(1 + \frac{\varepsilon}{2}\right)$-*approximation to the cost of the optimal low-rank approximation and achieves recourse* $\mathcal{O}\left(n\sqrt{k} + \frac{k}{\varepsilon}\log^2(ndM)\right)$.

---

**Algorithm 3** Relative-error algorithm for low-rank approximation with recourse $\frac{k^{3/2}}{\varepsilon^2} \cdot$ polylog$(ndM)$

**Input:** Rows $\mathbf{a}_1, \ldots, \mathbf{a}_n$ of input matrix $\mathbf{A} \in \mathbb{R}^{n \times d}$ with integer entries magnitude at most $M$
**Output:** $\left(1 + \frac{\varepsilon}{4}\right)$-approximation to the cost of the optimal low-rank approximation at all times
 1: **for** each row $\mathbf{a}_t$ **do**
 2:     Sample $\mathbf{a}_t$ with online ridge leverage score                         ▷Theorem 2.4
 3:     Run Algorithm 2 on the stream induced by the sampled rows
 4: **end for**

---

To reduce the number of rows in the input, we again apply Theorem 2.4, which, combined with Lemma 3.10, gives our main result Theorem 1.3 for Algorithm 3. Finally, we remark that since Theorem 2.4 samples $\mathcal{O}\left(\frac{k}{\varepsilon^2}\log n \log^2 \kappa\right)$ rows and there are input sparsity algorithms for approximations of the sampling probabilities (Cohen et al., 2017), then Algorithm 3 can be implemented used $\frac{kd}{\varepsilon} \cdot$ polylog$(ndM)$ bits of space and $d \cdot$ poly $\left(k, \frac{1}{\varepsilon}, \log(ndM)\right)$ amortized update time.

## 4 EMPIRICAL EVALUATIONS

We describe our empirical evaluations on a large-scale real-world dataset, comparing the quality of the solution of our algorithm to the quality of the optimal low-rank approximation solution. We discuss a number of additional experiments on both synthetic and real-world datasets in Appendix G. All experiments were conducted utilizing Python version 3.10.4 on a 64-bit operating system running on an AMD Ryzen 7 5700U CPU. The system was equipped with 8GB of RAM and featured 8 cores, each operating at a base clock speed of 1.80 GHz. The code is publicly accessible at https://github.com/samsonzhou/consistent-LRA.

| $k$ | $(1+\varepsilon)$ | Median | Std. Dev. | Mean |
|---|---|---|---|---|
| | 1.1 | 1.000 | 0.0000 | 1.0000 |
| | 2 | 1.000 | 0.0000 | 1.0000 |
| 25 | 5 | 1.0000 | 0.0367 | 1.0016 |
| | 10 | 1.0000 | 2.4463 | 1.598 |
| | 100 | 1.1907 | 51.9353 | 7.8882 |

Table 1: Median, standard deviation, and mean for ratios of cost across various values of accuracy parameters for landmark dataset, between 150 and 5000 updates

**Experimental setup.** In this section, we focus on our evaluations Algorithm 4 on the Landmark dataset from the SuiteSparse Matrix Collection (Davis & Hu, 2011), which is commonly used in benchmark comparison for low-rank approximation, e.g., (Ban et al., 2019). The dataset consists of a total of 71952 rows with $d = 2704$ features. As our theoretical results prove that our algorithm has a small amount of recourse, we first compare the cost of the solution output by Algorithm 4 with the cost of the optimal low-rank approximation. However, determining the optimal cost over each time is computationally expensive and serves as the main bottleneck. Specifically, for a stream of length $n$, the baseline requires $n \cdot \mathcal{O}(n^\omega) = \Omega(n^3)$ runtime for $n \approx d$, where $\omega \approx 2.37$ is the exponent for matrix multiplication (Alman et al., 2025). Thus we consider the first $n = 5000$ rows for our data stream, so the goal was to perform low-rank approximation on every single prefix matrix of size $n' \times d$ with $n' \leq n$. In particular, we computed in the runtimes and ratios of the two costs for $k = 25$ across $c = (1 + \varepsilon) \in \{1.1, 2.5, 5, 10, 100\}$ in Figure 2 with central statistics in Table 1,

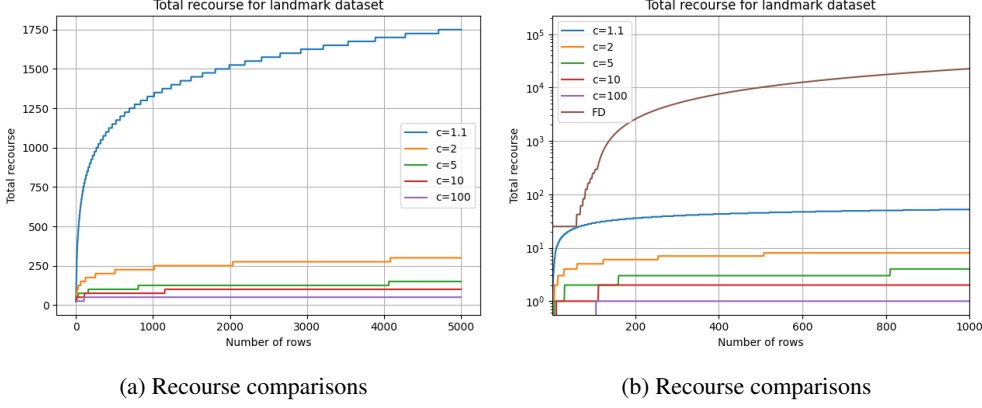

(a) Recourse comparisons

(b) Recourse comparisons

Fig. 1: Recourse comparisons for $k = 25$, $c = (1 + \varepsilon) \in \{1.1, 2.5, 5, 10, 100\}$

even though Algorithm 4 only guarantees additive error, rather than multiplicative error. Finally, we compared the recourse of our algorithms in Figure 1, along with FREQUENTDIRECTIONS, labeled FD, a standard algorithm for online low-rank approximation (Ghashami et al., 2016).

**Results and discussion.** Our results show a strong separation in the quality of online low-rank approximations such as Frequent Directions and our algorithms, which were specifically designed to achieve low recourse. Namely, for $n = 5000$, Frequent Directions has achieved recourse $121904$ while our algorithms range from recourse $100$ to $300$, more than a factor of $400X$. Moreover, our results show that the approximation guarantees of our algorithms are actually quite good in practice, especially as the number of rows increases; we believe the large variance in Figure 2b is due to the optimal low-rank approximation cost being quite small compared to the additive Frobenius error. Thus it seems our empirical evaluations provide compelling evidence that our algorithms achieve significantly better recourse than existing algorithms for online low-rank approximation; we provide a number of additional experiments in Appendix G.

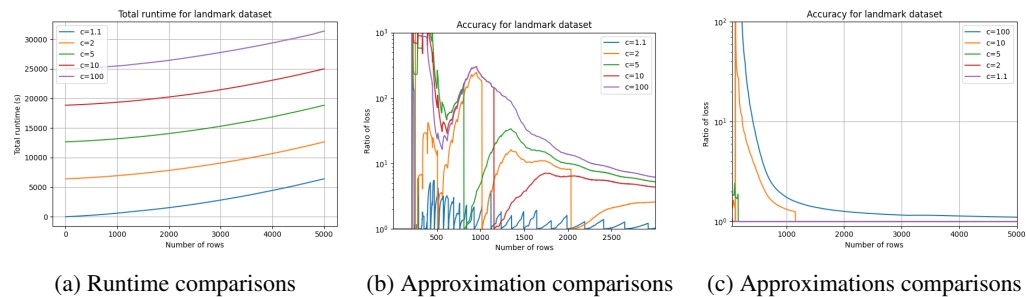

(a) Runtime comparisons

(b) Approximation comparisons

(c) Approximations comparisons

Fig. 2: Runtime and approximations on landmark dataset, for $k = 25$, $c = (1 + \varepsilon) \in \{1.1, 2.5, 5, 10, 100\}$

### ACKNOWLEDGMENTS

David P. Woodruff is supported in part Office of Naval Research award number N000142112647, and a Simons Investigator Award. Samson Zhou is supported in part by NSF CCF-2335411. Samson Zhou gratefully acknowledges funding provided by the Oak Ridge Associated Universities (ORAU) Ralph E. Powe Junior Faculty Enhancement Award.

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

## A  PRELIMINARIES

We use $[n]$ to denote the set $\{1, \ldots, n\}$. We use $\text{poly}(n)$ to denote a fixed polynomial in $n$, which can be adjusted using constants in the parameter settings. We use $\text{polylog}(n)$ to denote $\text{poly}(\log n)$. We use $\circ$ to denote the vertical concatenation of rows $\mathbf{a}_1, \mathbf{a}_2 \in \mathbb{R}^d$, so that $\mathbf{a}_1 \circ \mathbf{a}_2 = \begin{bmatrix} \mathbf{a}_1 \\ \mathbf{a}_2 \end{bmatrix}$. We say an event $\mathcal{E}$ occurs with high probability if $\mathbf{Pr}\,[\mathcal{E}] \geq 1 - \frac{1}{\text{poly}(n)}$. Recall that the Frobenius norm of a matrix $\mathbf{A} \in \mathbb{R}^{n \times d}$ is defined by $\|\mathbf{A}\|_F = \left( \sum_{i=1}^n \sum_{j=1}^d A_{i,j}^2 \right)^{1/2}$.

The singular value decomposition of a matrix $\mathbf{A} \in \mathbb{R}^{n \times d}$ with rank $r$ is the decomposition $\mathbf{A} = \mathbf{U}\mathbf{\Sigma}\mathbf{V}$ for $\mathbf{U} \in \mathbb{R}^{n \times r}$, $\mathbf{\Sigma} \in \mathbb{R}^{r \times r}$, $\mathbf{V} \in \mathbb{R}^{r \times d}$, where the columns of $\mathbf{U}$ are orthonormal, the rows of $\mathbf{V}$ are orthonormal, and $\mathbf{\Sigma}$ is a diagonal matrix whose entries correspond to the singular values of $\mathbf{A}$.

**Lemma A.1.** *For subspaces $\mathbf{R}$ and $\mathbf{T}$ of rank $k$, let their corresponding orthogonal projection matrices be $\mathbf{P}$ and $\mathbf{Q}$. Then there exist constants $C_1, C_2 > 0$ such that $C_1(\|\mathbf{P} - \mathbf{PQ}\|_F^2 + \|\mathbf{Q} - \mathbf{QP}\|_F^2) \leq \text{Recourse}(\mathbf{R}, \mathbf{T}) \leq C_2(\|\mathbf{P} - \mathbf{PQ}\|_F^2 + \|\mathbf{Q} - \mathbf{QP}\|_F^2).$*

*Proof.* By definition, we have $\text{Recourse}(\mathbf{R}, \mathbf{T}) = \|\mathbf{P} - \mathbf{Q}\|_F^2$. By the triangle inequality, we have $\|\mathbf{P} - \mathbf{Q}\|_F \leq \|\mathbf{P} - \mathbf{PQ}\|_F + \|\mathbf{PQ} - \mathbf{Q}\|_F$. Observe that $\|\mathbf{Q} - \mathbf{QP}\|_F^2 = \|\mathbf{Q} - \mathbf{PQ}\|_F^2$ because the left-hand side is the trace of $(\mathbf{Q} - \mathbf{QP})^\top(\mathbf{Q} - \mathbf{QP})$, which equals the trace of $(\mathbf{Q} - \mathbf{PQ})(\mathbf{Q} - \mathbf{QP})$, since $\mathbf{P}^\top = \mathbf{P}$ and $\mathbf{Q}^\top = \mathbf{Q}$ for projection matrices $\mathbf{Q}$ and $\mathbf{P}$. By the cyclic property of trace, the trace of $(\mathbf{Q} - \mathbf{PQ})(\mathbf{Q} - \mathbf{QP})$ thus equals the trace of $(\mathbf{Q} - \mathbf{QP})(\mathbf{Q} - \mathbf{PQ})$, which by the same argument, is the trace of the right-hand side. Hence it follows that $\|\mathbf{Q} - \mathbf{QP}\|_F^2 = \|\mathbf{Q} - \mathbf{PQ}\|_F^2$, as desired. Thus, we have $\|\mathbf{P} - \mathbf{Q}\|_F \leq \|\mathbf{P} - \mathbf{PQ}\|_F + \|\mathbf{QP} - \mathbf{Q}\|_F$.

Next, observe that

$$\|\mathbf{P} - \mathbf{PQ}\|_F^2 + \|\mathbf{PQ} - \mathbf{Q}\|_F^2 = \text{Trace}(\mathbf{P}) + \text{Trace}(\mathbf{Q}) - 2\,\text{Trace}(\mathbf{PQ})$$
$$= 2k - 2\,\text{Trace}(\mathbf{PQ}),$$

since $\mathbf{P}^2 = \mathbf{P}$, $\mathbf{Q}^2 = \mathbf{Q}$ are symmetric idempotents, and $\text{Trace}(\mathbf{P}) = \text{Trace}(\mathbf{Q}) = k$. Similarly,

$$2k + 2\|\mathbf{QP}\|_F^2 - 4\,\text{Trace}(\mathbf{PQ}) = 2k + 2\,\text{Trace}(\mathbf{PQP}) - 4\,\text{Trace}(\mathbf{PQ})$$
$$= 2k - 2\,\text{Trace}(\mathbf{PQ}),$$

using $\|\mathbf{QP}\|_F^2 = \text{Trace}(\mathbf{PQP}) = \text{Trace}(\mathbf{PQ})$. Thus, we have $\|\mathbf{P} - \mathbf{PQ}\|_F^2 + \|\mathbf{PQ} - \mathbf{Q}\|_F^2 = 2k + 2\|\mathbf{QP}\|_F^2 - 4 \cdot \text{Trace}(\mathbf{P}^\top\mathbf{Q})$. The latter quantity is at most $4k - 4 \cdot \text{Trace}(\mathbf{P}^\top\mathbf{Q}) = 2(2k - 2 \cdot \text{Trace}(\mathbf{P}^\top\mathbf{Q}))$, which is just $2\|\mathbf{P} - \mathbf{Q}\|_F^2$. $\qquad\square$

Thus, it suffices to work with the less natural but perhaps more mathematically accessible definition of $\|\mathbf{R} - \mathbf{RT}^\dagger\mathbf{T}\|_F^2 + \|\mathbf{T} - \mathbf{TR}^\dagger\mathbf{R}\|_F^2$, i.e., the symmetric difference of the mass of the two subspaces, as a notion of recourse.

**Theorem A.2** (Min-max theorem). *Let $\mathbf{A} \in \mathbb{R}^{n \times d}$ be a matrix with singular values $\sigma_1(\mathbf{A}) \geq \sigma_2(\mathbf{A}) \geq \ldots \geq \sigma_d(\mathbf{A})$ and let $\xi_j(\mathbf{A}) = \sigma_{d-j+1}(\mathbf{A})$ for all $j \in [d]$, so that $\xi_1(\mathbf{A}) \leq \ldots \leq \xi_d(\mathbf{A})$ is the reverse spectrum of $\mathbf{A}$. Then for any subspace $\mathbf{V}$ of $\mathbf{A}$ with dimension $k$, there exist unit vectors $\mathbf{x}, \mathbf{y} \in \mathbf{V}$ such that*

$$\|\mathbf{V}\mathbf{x}\|_2^2 \leq \sigma_k^2(\mathbf{A}), \qquad \|\mathbf{V}\mathbf{y}\|_2^2 \geq \xi_k^2(\mathbf{A}).$$

**Theorem A.3** (Cauchy interlacing theorem). *(Hwang, 2004; Fisk, 2005) Let $\mathbf{A} \in \mathbb{R}^{n \times d}$ be a matrix with singular values $\sigma_1(\mathbf{A}) \geq \sigma_2(\mathbf{A}) \geq \ldots \geq \sigma_d(\mathbf{A})$. Let $\mathbf{v} \in \mathbb{R}^d$ and $\mathbf{B} = \mathbf{A} \circ \mathbf{v} \in \mathbb{R}^{(n+1) \times d}$ with singular values $\sigma_1(\mathbf{B}) \geq \sigma_2(\mathbf{B}) \geq \ldots \geq \sigma_d(\mathbf{B})$. Then $\sigma_i(\mathbf{B}) \geq \sigma_i(\mathbf{A})$ for all $i \in [d]$.*

## B  TECHNICAL OVERVIEW

In this section, we provide intuition for our main results, summarizing our algorithms, various challenges, as well as natural other approaches and why they do not work.

**Warm-up: additive error algorithm.** As a simple warm-up, we first describe our algorithm that achieves additive error at most $\varepsilon \cdot \|\mathbf{A}^{(t)}\|_F^2$ across all times $t \in [n]$, while only incurring recourse $\mathcal{O}\left(\frac{k}{\varepsilon} \log(ndM)\right)$, corresponding to Theorem 1.1. This algorithm is quite simple. We maintain the squared Frobenius norm of the matrix $\mathbf{A}^{(t)}$ across all times $t \in [n]$. We also maintain the same set of factors provided the squared Frobenius norm has not increased by a factor of $(1 + \varepsilon)$ since the previous time we changed the set of factors. When the squared Frobenius norm has increased by a factor of $(1+\varepsilon)$ at a time $t$ since the previous time we changed the set of factors, then we simply use the singular value decomposition of $\mathbf{A}^{(t)}$ to set $\mathbf{V}^{(t)}$ to be the top $k$ right singular vectors of $\mathbf{A}^{(t)}$. Since we change the entire set of factors, this process can incur recourse cost at most $k$.

The correctness follows from the observation that each time we reset $\mathbf{V}^{(t)}$, we find the optimal solution at time $t$. Now, over the next few times after $t$, our solution can only degrade by the squared Frobenius norm of the submatrix formed by the incoming rows, which is less than an $\varepsilon$-fraction of the squared Frobenius norm of the entire matrix, due to the requirement that we recompute $\mathbf{V}^{(t)}$ each time the squared Frobenius norm has increased by a $(1 + \varepsilon)$ factor. Since the Frobenius norm can only increase by a multiplicative $(1+\varepsilon)$ factor a total of at most $\mathcal{O}\left(\frac{1}{\varepsilon} \log(ndM)\right)$ times and we incur recourse cost $k$ each time, then the resulting recourse is at most the desired $\mathcal{O}\left(\frac{k}{\varepsilon} \log(ndM)\right)$.

**Stream reduction for relative error algorithm.** We now discuss the goal of achieving $\frac{k^{3/2}}{\varepsilon^2} \cdot \text{polylog}(ndM)$ recourse while maintaining a relative $(1 + \varepsilon)$-multiplicative approximation to the optimal low-rank approximation cost at all times, i.e., Theorem 1.3. We first utilize online ridge-leverage score sampling (Braverman et al., 2020) to sample $\frac{k}{\varepsilon} \cdot \text{polylog}(ndM)$ rows of the stream $\mathcal{S}$ of rows $\mathbf{a}_1, \ldots, \mathbf{a}_n$ on-the-fly, to form a stream $\mathcal{S}'$ consisting of reweighted rows $\mathbf{b}_1, \ldots, \mathbf{b}_m$ of $\mathbf{A}$ with $m = \frac{k}{\varepsilon} \cdot \text{polylog}(ndM)$. By the guarantees of online ridge-leverage score sampling, to achieve a $(1 + \varepsilon)$-approximation to the matrix $\mathbf{A}^{(t)}$ consisting of the rows $\mathbf{a}_1, \ldots, \mathbf{a}_t$, it suffices to achieve a $(1 + \mathcal{O}(\varepsilon))$-approximation to the matrix $\mathbf{B}^{(t')}$ consisting of the rows $\mathbf{b}_1, \ldots, \mathbf{b}_{t'}$ that have arrived by time $t$. Note that since $\mathbf{B}$ is a submatrix of $\mathbf{A}$, we have $t' \le t$. Moreover, the rows of $\mathbf{A}$ that are sampled into $\mathbf{B}$ are only increased by at most a $\text{poly}(n)$ factor, so we can assume that the magnitude of the entries is still bounded polynomially by $n$. Thus it suffices to perform consistent low-rank approximation on the matrix $\mathbf{B}$ instead. Hence for the remainder of the discussion, we assume that the stream length is $\frac{k}{\varepsilon} \cdot \text{polylog}(ndM)$ rather than $n$.

**Relative error algorithm on reduced stream.** We now describe our algorithm for $(1 + \varepsilon)$-multiplicative relative error at all times of the stream of length $\frac{k}{\varepsilon} \cdot \text{polylog}(ndM)$. At some time $s$ in the stream, we use the singular value decomposition of $\mathbf{A}^{(s)}$ to compute the top $k$ right singular values of $\mathbf{A}^{(s)}$, which we then set to be our factor $\mathbf{V}^{(s)}$. We can do this each time the optimal low-rank approximation cost has increased by a multiplicative $(1 + \mathcal{O}(\varepsilon))$-factor since the previous time we set our factor to be $\mathbf{V}^{(s)}$. We first discuss how to maintain a $(1 + \varepsilon)$-approximation with the desired recourse in between the times at which the optimal low-rank approximation cost has increased by a $(1 + \varepsilon)$-multiplicative factor.

Towards this goal, we perform casework on the contribution of the bottom $\sqrt{k}$ singular values within the top $k$ right singular values of $\mathbf{V}^{(s)}$. Namely, if $\sum_{i=k-\sqrt{k}}^{k} \sigma_i^2(\mathbf{A}^{(s)})$ is "small", then intuitively, the corresponding singular vectors can be replaced without substantially increasing the error. Hence in this case, we can replace one of the bottom $\sqrt{k}$ singular vectors with the new row each time a new incoming row arrives. This procedure will incur $\sqrt{k}$ total recourse over the next $\sqrt{k}$ updates, after which at time $t$ we reset the factor to be the top $k$ right singular vectors of the matrix $\mathbf{A}^{(t)}$. Therefore, we incur recourse $\mathcal{O}(k)$ across $\sqrt{k}$ time steps.

On the other hand, if $\sum_{i=k-\sqrt{k}}^{k} \sigma_i^2(\mathbf{A}^{(s)})$ is "large", then the optimal low-rank approximation factors cannot be projected away too much from these directions, since otherwise the optimal low-rank approximation cost would significantly increase. Thus, we can simply choose the optimal set of top right $k$ singular vectors at each time, because there will be substantial overlap between the new subspace and the old subspace. In particular, if we choose our threshold to be $\sum_{i=k-\sqrt{k}}^{k} \sigma_i^2(\mathbf{A}^{(s)})$ to be an $\mathcal{O}(\varepsilon)$-factor of the optimal cost, then we show that incurring $\sqrt{k}$ recourse will increase the low-rank approximation cost by $(1 + \varepsilon)$, which violates the assumption that we consider times

during which the optimal low-rank approximation cost has not increased by a $(1+\varepsilon)$-multiplicative factor. Therefore, the recourse is at most $\sqrt{k}$ across each time.

In summary, between the times at which the optimal low-rank approximation cost has increased by a $(1+\varepsilon)$-multiplicative factor, we incur at most $\sqrt{k}$ recourse for each time. Since the stream length is $\frac{k}{\varepsilon} \cdot \mathrm{polylog}(ndM)$, we then incur $\frac{k^{1.5}}{\varepsilon} \cdot \mathrm{polylog}(ndM)$ total recourse across these times, which is our desired bound. It remains to bound the total recourse at times $s$ when the optimal low-rank approximation cost has increased by a $(1+\varepsilon)$ multiplicative factor, as we reset our solution to be the top $k$ right singular values, incurring recourse $k$ at each of these times. Because the Frobenius norm is at most $\mathrm{poly}(ndM)$, a natural conclusion would be that the optimal low-rank approximation cost can increase by a $(1+\varepsilon)$ multiplicative factor at most $\mathcal{O}\left(\frac{1}{\varepsilon} \log(ndM)\right)$ times. Unfortunately, this is not the case.

**Anti-Hadamard matrices.** Problematically, there exist constructions of anti-Hadamard integer matrices, which have dimension $n \times d$ but optimal low-rank cost as small as $\exp(-\mathcal{O}(k))$. Hence, the optimal cost can double $\mathcal{O}(k)$ times, thereby incurring total recourse $\mathcal{O}\left(k^{2.5}\right)$, which is undesirably large. Instead, we show that when the optimal low-rank cost is exponentially small, then the rank of the matrix must also be quite small, meaning that the recourse of our algorithm cannot be as large as in the full-rank case. Namely, we generalize a result by Clarkson & Woodruff (2009) to show that if an integer matrix $\mathbf{A} \in \mathbb{Z}^{n \times d}$ has rank $r > k$ and entries bounded in magnitude by $M$, then its optimal low-rank approximation cost is at least $(ndM^2)^{-\frac{k}{r-k}}$. Hence, we only need to consider anti-Hadamard matrices when the rank is less than $2k$.

Fortunately, when the rank is at most $r < 2k$, we can apply a more fine-grained analysis for the above cases, since there are only $r$ vectors spanning the row span, so the recourse in many of the previous operations can be at most $r - k$. In particular, for $r < k$, we can simply maintain the entire row span. We then argue that if the rank $r$ of the matrix is between $k + 2^i$ and $k + 2^{i+1}$, then there can be at most $\mathcal{O}\left(\frac{1}{\varepsilon} \frac{k}{2^i}\right)$ epochs before the cost of the optimal low-rank approximation is at least $(ndM^2)^{-100}$. Moreover, the recourse incurred by recomputing the top eigenspace is at most $r - k \leq 2^{i+1}$, so that the total recourse for the times where the rank of the matrix is between $k + 2^i$ and $k + 2^{i+1}$ is at most $\mathcal{O}\left(\frac{k}{\varepsilon} \log(ndM)\right)$. It then follows that the total recourse across all times before the rank becomes at least $2k$ is at most $\mathcal{O}\left(\frac{k}{\varepsilon} \log^2(ndM)\right)$.

A keen reader might ask whether $k^{3/2}$ is best possible recourse for our algorithmic approach. To that end, observe that if we change a large number of factors per update, then the total recourse will increase. Let us suppose that there are roughly $k$ updates, which is the size of the online projection-cost preserving coreset. Now if we change $r$ factors per update, then the total recourse will be at least $kr$. On the other hand, if we change a smaller number of factors per update, then we will need to recompute every $\frac{k}{r}$ steps. Each recompute takes $k$ recourse, for a total of $\frac{k^2}{r}$ overall recourse. Hence, the maximum of the quantities $kr$ and $\frac{k^2}{r}$ is minimized at $r = \sqrt{k}$, giving recourse $kr = \frac{k^2}{r} = k^{3/2}$.

**Recourse lower bound.** Our lower bound construction that shows recourse $\Omega\left(\frac{k}{\varepsilon} \log \frac{n}{k}\right)$ is necessary, corresponding to Theorem 1.4, is simple. We divide the stream into $\Theta\left(\frac{1}{\varepsilon} \log \frac{n}{k}\right)$ phases where the optimal low-rank approximation cost increases by a multiplicative $(1 + \mathcal{O}(\varepsilon))$ factor between each phase. Moreover, the optimal solution to the $i$-th phase is orthogonal to the optimal solution to the $(i - 1)$-th phase, so that depending on the parity of the phase $i$, the optimal solution is either the first $k$ elementary vectors, or the elementary vectors $k + 1$ through $2k$. Therefore, a multiplicative $(1+\varepsilon)$-approximation at all times requires incurring $\Omega(k)$ recourse between each phase, which shows the desired $\Omega\left(\frac{k}{\varepsilon} \log \frac{n}{k}\right)$ lower bound.

## C  ADDITIONAL RELATED WORK

We remark that there has been a flurry of recent work studying consistency for various problems. The problem of consistent clustering was initialized by Lattanzi & Vassilvitskii (2017), who gave an algorithm with recourse $k^2 \cdot \mathrm{polylog}(n)$ for $k$-clustering on insertion-only streams, i.e., the incremental setting. This recourse bound was subsequently improved to $k \cdot \mathrm{polylog}(n)$ by Fichtenberger

et al. (2021), while a version robust to outliers was presented by Guo et al. (2021). The approximation guarantee was also recently improved to $(1 + \varepsilon)$ by Chan et al. (2025). A line of recent work has studied $k$-clustering in the dynamic setting Lacki ⓡ et al. (2023); Bhattacharya et al. (2024); Forster & Skarlatos (2025), where points may be inserted and deleted. Rather than $k$-clustering, Cohen-Addad et al. (2022) studied consistency for correlation clustering, where edges are positively or negatively labeled, and the goal is to form as many clusters as necessary to minimize the number of negatively labeled edges within a cluster and the number of positively labeled edges between two different clusters. For problems beyond clustering, a line of work has also focused on submodular maximization (Jaghargh et al., 2019; Duetting et al., 2024; Dütting et al., 2024).

**Consistent clustering.** Another approach might be to adapt ideas from the consistent clustering literature. In this setting, a sequence of points in $\mathbb{R}^d$ arrive one-by-one, and the goal is to maintain a constant-factor approximation to the $(k, z)$-clustering cost, while minimizing the total recourse. Here, the recourse incurred at a time $t$ is the size of the symmetric difference between the clustering centers selected at time $t - 1$ and at time $t$. The only algorithm to achieve recourse subquadratic in $k$ is the algorithm by Fichtenberger et al. (2021), which attempts to create robust clusters at each time by looking at geometric balls with increasing radius around each existing point to pick centers that are less sensitive to possible future points. Unfortunately, such a technique utilizes the geometric properties implicit in the objective of $k$-clustering and it is not obvious what the corresponding analogues should be for low-rank approximation.

## D RECOURSE LOWER BOUND

In this section, we prove our recourse lower bound. The main idea is to simply partition the data stream into $\Theta\left(\frac{1}{\varepsilon} \log \frac{n}{k}\right)$ phases, so that the optimal low-rank approximation cost increases by a multiplicative $(1 + \mathcal{O}(\varepsilon))$-factor between each phase. We also design the input matrix so that the optimal solution to the $i$-th phase is orthogonal to the optimal solution to the $(i-1)$-th phase. Hence, depending on the parity of the phase $i$, the optimal solution is either the first $k$ elementary vectors, or the elementary vectors $k + 1$ through $2k$ and thus a $(1 + \varepsilon)$-approximation at all times requires incurring $\Omega(k)$ recourse between each phase, which shows the desired $\Omega\left(\frac{k}{\varepsilon} \log \frac{n}{k}\right)$ lower bound.

**Theorem 1.4.** *For any parameter $\varepsilon > \frac{\log n}{n}$, there exists a sequence of rows $\mathbf{x}_1, \ldots, \mathbf{x}_n \in \mathbb{R}^d$ such that any algorithm that produces a $(1 + \varepsilon)$-approximation to the cost of the optimal low-rank approximation at all times must have consistency cost $\Omega\left(\frac{k}{\varepsilon} \log \frac{n}{k}\right)$.*

*Proof.* We divide the stream into $\Theta\left(\frac{1}{\varepsilon} \log \frac{n}{k}\right)$ phases. Let $C > 2$ be some parameter that we shall set. If $i$ is odd, then in the $i$-th phase, we add $(1 + \varepsilon)^i$ copies of the elementary vectors $\mathbf{e}_1, \ldots, \mathbf{e}_k$. If $i$ is even, then in the $i$-th phase, we add $(1 + C\varepsilon)^i$ copies of the elementary vectors $\mathbf{e}_{k+1}, \ldots, \mathbf{e}_{2k}$. Note that since there are $\Theta\left(\frac{1}{\varepsilon} \log \frac{n}{k}\right)$ phases and each phase inserts $(1 + C\varepsilon)^i$ copies of $k$ rows, then the total number of copies of each row inserted is at most $\mathcal{O}(n) \, 2k$ for the correct fixing of the constant in the $\Theta(\cdot)$ notation, and thus there are at most $n$ rows overall.

We remark that by construction, after the $i$-th phase, the optimal rank-$k$ approximation is the elementary vectors $\mathbf{e}_1, \ldots, \mathbf{e}_k$ if $i$ is odd and the elementary vectors $\mathbf{e}_{k+1}, \ldots, \mathbf{e}_{2k}$ if $i$ is even. In particular, by the Eckart-Young-Mirsky theorem, i.e., Theorem 3.1, after the $i$-th phase, the optimal rank-$k$ approximation to the underlying matrix induces cost $k \cdot \left((1 + C\varepsilon)^2 + (1 + C\varepsilon)^4 + \ldots + (1 + C\varepsilon)^{i-1}\right)$ if $i$ is odd and $k \cdot \left((1 + C\varepsilon) + (1 + C\varepsilon)^3 + \ldots + (1 + C\varepsilon)^{i-1}\right)$ if $i$ is even. Note that both of these quantities are at most $2k(1 + C\varepsilon)^{i-1}$. Thus for the time $t$ after an odd phase $i$, a matrix $\mathbf{M}$ of rank-$k$ factors must satisfy

$$\|\mathbf{X}^{(t)} - \mathbf{X}^{(t)}\mathbf{M}^{\dagger}\mathbf{M}\|_F^2 \leq 2\varepsilon k(1 + C\varepsilon)^{i-1},$$

in order to be a $(1 + \varepsilon)$-approximation to the optimal low-rank cost.

Let $\mathbf{E}^{(1)} = \mathbf{e}_1 \circ \ldots \circ \mathbf{e}_k$ and $\mathbf{E}^{(2)} = \mathbf{e}_{k+1} \circ \ldots \circ \mathbf{e}_{2k}$. For any constant $C > 100$, it follows that $\mathbf{M}$ must have squared mass at least $k(1 - 2\varepsilon)$ onto $\mathbf{E}^{(1)}$ to be a $(1 + \varepsilon)$-approximation to the cost of the optimal low-rank approximation to $\mathbf{X}^{(t)}$, i.e., $\|\mathbf{M}(\mathbf{E}^{(1)})^{\top}\mathbf{E}^{(1)}\|_F^2 \geq k(1 - 2\varepsilon)$ for the time $t$ immediately following an odd phase $i$. By similar reasoning, $\mathbf{M}$ must have squared mass at least

$k(1-2\varepsilon)$ onto $\mathbf{E}^{(2)}$ to be a $(1+\varepsilon)$-approximation to the cost of the optimal low-rank approximation to $\mathbf{X}^{(t)}$ for the time $t$ immediately following an odd phase $i$. However, because $\mathbf{E}^{(1)}$ and $\mathbf{E}^{(2)}$ are disjoint, then it follows that $\mathbf{M}$ must have recourse $\Omega(k)$ between each phase. Since there are $\Omega\left(\frac{1}{\varepsilon}\log\frac{n}{k}\right)$ phases, then the total recourse must be $\Omega\left(\frac{k}{\varepsilon}\log\frac{n}{k}\right)$. □

## E    MISSING PROOFS FROM SECTION 2

---

**Algorithm 4** Additive error algorithm for low-rank approximation with low recourse

---

**Input:** Rows $\mathbf{a}_1, \ldots, \mathbf{a}_n$ of input matrix $\mathbf{A} \in \mathbb{R}^{n \times d}$ with integer entries bounded in magnitude by $M$, error parameter $\varepsilon > 0$
**Output:** Additive $\varepsilon \cdot \|\mathbf{A}^{(t)}\|_F^2$ error to the cost of the optimal low-rank approximation at all times
 1: $C \leftarrow 0$
 2: **for** each row $\mathbf{a}_t$ **do**
 3:     **if** $\|\mathbf{A}^{(t)}\|_F^2 \geq (1+\varepsilon) \cdot C$ **then**
 4:         $\mathbf{V} \leftarrow \text{RECLUSTER}(\mathbf{A}^{(t)}, k)$
 5:         $C \leftarrow \|\mathbf{A}^{(t)}\|_F^2$
 6:     **end if**
 7:     Return $\mathbf{V}$
 8: **end for**

---

We show correctness of Algorithm 4 at all times:

**Lemma E.1.** *Let $\mathbf{A}^{(t)}$ be the first $t$ rows of $\mathbf{A}$ and let $\mathbf{V}^{(t)}$ be the output of Algorithm 4 at time $t$. Let $\mathsf{OPT}_t$ be the cost of the optimal low-rank approximation at time $t$. Then $\|\mathbf{A}^{(t)} - \mathbf{A}^{(t)}(\mathbf{V}^{(t)})^\top \mathbf{V}^{(t)}\|_F^2 \leq \mathsf{OPT}_t + \varepsilon \cdot \|\mathbf{A}^{(t)}\|_F^2$.*

It then remains to bound the recourse of Algorithm 4:

**Lemma E.2.** *The recourse of Algorithm 4 is at most $\mathcal{O}\left(\frac{k}{\varepsilon}\log(ndM)\right)$.*

Theorem 1.1 then follows from Lemma E.1 and Lemma E.2.

**Lemma E.1.** *Let $\mathbf{A}^{(t)}$ be the first $t$ rows of $\mathbf{A}$ and let $\mathbf{V}^{(t)}$ be the output of Algorithm 4 at time $t$. Let $\mathsf{OPT}_t$ be the cost of the optimal low-rank approximation at time $t$. Then $\|\mathbf{A}^{(t)} - \mathbf{A}^{(t)}(\mathbf{V}^{(t)})^\top \mathbf{V}^{(t)}\|_F^2 \leq \mathsf{OPT}_t + \varepsilon \cdot \|\mathbf{A}^{(t)}\|_F^2$.*

*Proof.* Let $s$ be the time at which $\mathbf{V}^{(t-1)}$ was first set, so that $\mathbf{V}^{(t-1)} = \mathbf{V}^{(s)}$ are the top $k$ right singular vectors of $\mathbf{A}^{(s)}$. Therefore, $\|\mathbf{A}^{(s)} - \mathbf{A}^{(s)}(\mathbf{V}^{(s)})^\top \mathbf{V}^{(s)}\|_F^2 = \mathsf{OPT}_s$. Hence,

$$\|\mathbf{A}^{(t)} - \mathbf{A}^{(t)}(\mathbf{V}^{(t)})^\top \mathbf{V}^{(t)}\|_F^2 = \|\mathbf{A}^{(s)} - \mathbf{A}^{(s)}(\mathbf{V}^{(t)})^\top \mathbf{V}^{(t)}\|_F^2 + \sum_{i=s+1}^{t} \|\mathbf{a}_i - \mathbf{a}_i(\mathbf{V}^{(t)})^\top \mathbf{V}^{(t)}\|_F^2$$

$$= \|\mathbf{A}^{(s)} - \mathbf{A}^{(s)}(\mathbf{V}^{(s)})^\top \mathbf{V}^{(s)}\|_F^2 + \sum_{i=s+1}^{t} \|\mathbf{a}_i - \mathbf{a}_i(\mathbf{V}^{(t)})^\top \mathbf{V}^{(t)}\|_F^2$$

$$= \mathsf{OPT}_s + \sum_{i=s+1}^{t} \|\mathbf{a}_i - \mathbf{a}_i(\mathbf{V}^{(t)})^\top \mathbf{V}^{(t)}\|_2^2.$$

Note that since $(\mathbf{V}^{(t)})^\top \mathbf{V}^{(t)}$ is a projection operator, then the length of $\mathbf{a}_i$ cannot increase after being projected onto the row span of $\mathbf{V}^{(t)}$, so that $\|\mathbf{a}_i - \mathbf{a}_i(\mathbf{V}^{(t)})^\top \mathbf{V}^{(t)}\|_2^2 \leq \|\mathbf{a}_i\|_2^2$. Therefore,

$$\|\mathbf{A}^{(t)} - \mathbf{A}^{(t)}(\mathbf{V}^{(t)})^\top \mathbf{V}^{(t)}\|_F^2 = \mathsf{OPT}_s + \sum_{i=s+1}^{t} \|\mathbf{a}_i - \mathbf{a}_i(\mathbf{V}^{(t)})^\top \mathbf{V}^{(t)}\|_2^2$$

$$\leq \mathsf{OPT}_s + \sum_{i=s+1}^{t} \|\mathbf{a}_i\|_2^2$$

$$\leq \mathsf{OPT}_s + \varepsilon \cdot \|\mathbf{A}^{(s)}\|_F^2,$$

where the last inequality is due to Line 3 of Algorithm 4. Finally, by the monotonicity of the optimal low-rank approximation cost with additional rows, we have that $\mathsf{OPT}_s \leq \mathsf{OPT}_t$ and thus,

$$\|\mathbf{A}^{(t)} - \mathbf{A}^{(t)}(\mathbf{V}^{(t)})^\top \mathbf{V}^{(t)}\|_F^2 \leq \mathsf{OPT}_t + \varepsilon \cdot \|\mathbf{A}^{(s)}\|_F^2,$$

as desired. □

**Lemma E.2.** *The recourse of Algorithm 4 is at most $\mathcal{O}\left(\frac{k}{\varepsilon}\log(ndM)\right)$.*

*Proof.* Since each entry of $\mathbf{A}$ is an integer bounded in magnitude by at most $M$, then the squared Frobenius norm of $\mathbf{A}$ is at most $(nd)M^2$. Moreover, each entry of $\mathbf{A}$ is an integer bounded, the first time it is nonzero, the squared Frobenius norm must be at least 1. Hence, the squared Frobneius norm of $\mathbf{A}$ can increase by a factor of $(1 + \varepsilon)$ at most $\log_{(1+\varepsilon)}(nd)M^2 = \mathcal{O}\left(\frac{1}{\varepsilon}\log(ndM)\right)$ from the first time it is nonzero. Each time $t$ it does so, we recompute the right singular values of $\mathbf{A}^{(t)}$ to be the set of factors $\mathbf{V}^{(t)}$. Thus the recourse incurred at these times is at most $\mathcal{O}\left(\frac{k}{\varepsilon}\log(ndM)\right)$. For all other times, we retain the same choice of the factors. Hence the desired claim follows. □

We now show that the optimal rank $k$ subspace incurs at most constant recourse under rank one perturbations.

**Lemma 2.1.** *Let $\mathbf{A}^{(t-1)} \in \mathbb{R}^{(t-1)\times d}$ and $\mathbf{A}^{(t)} \in \mathbb{R}^{t\times d}$ such that $\mathbf{A}^{(t)}$ is $\mathbf{A}^{(t-1)}$ with the row $\mathbf{A}_t$ appended. Let $\mathbf{V}_{t-1}^*$ and $\mathbf{V}_t^*$ be the optimal rank-$k$ subspaces (the span of the top $k$ right singular vectors) of $\mathbf{A}^{(t-1)}$ and $\mathbf{A}^{(t)}$, respectively. Then $\mathrm{Recourse}(\mathbf{V}_{t-1}^*, \mathbf{V}_t^*) \leq 8$.*

*Proof.* The recourse between two subspaces is defined as the squared Frobenius norm of the difference between their corresponding orthogonal projection matrices. We first consider the covariance matrices induced by the optimal rank-$k$ subspaces. Namely, consider the covariance matrices $\mathbf{B}_{t-1} = (\mathbf{A}_{(t-1)}^*)^\top \mathbf{A}_{(t-1)}^*$ and $\mathbf{B}_t = (\mathbf{A}^{(t)})^\top \mathbf{A}^{(t)}$. Then we have the relationship $\mathbf{B}_t = \mathbf{B}_{t-1} + \mathbf{A}_t^\top \mathbf{A}$, i.e., $\mathbf{B}_t$ is obtained from $\mathbf{B}_{t-1}$ by a rank-1 positive semi-definite (PSD) update. By the Eckart-Young theorem, c.f., Theorem 3.1, the subspace $\mathbf{V}_t^*$ is the span of the top $k$ eigenvectors of $\mathbf{B}_t$, and similarly for $\mathbf{V}_{t-1}^*$ and $\mathbf{B}_{t-1}$.

We next consider the intersection of the eigenspaces of $\mathbf{B}_t$ and $\mathbf{B}_{t-1}$. We first aim to show that the dimension of the intersection of the two subspaces is at least $k - 1$, i.e., $\dim(\mathbf{V}_{t-1}^* \cap \mathbf{V}_t^*) \geq k - 1$. Let $\mathbf{S}_a$ be the subspace orthogonal to $\mathbf{A}_t$, so that $\mathbf{S}_a = \{\mathbf{v} \in \mathbb{R}^d : \mathbf{A}_t \mathbf{v} = 0\}$ and $\dim(\mathbf{S}_a) = d - 1$. Consider the intersection $\mathbf{W} = \mathbf{V}_{t-1}^* \cap \mathbf{S}_a$. By the properties of subspace dimensions:

$$\dim(\mathbf{W}) = \dim(\mathbf{V}_{t-1}^*) + \dim(\mathbf{S}_a) - \dim(\mathbf{V}^{t-1} \cap \mathbf{S}_a).$$

Since $\dim(b\mathbf{V}_{t-1}^*) = k$, $\dim(\mathbf{S}_a) = d - 1$, and $\dim(\mathbf{V}_{t-1}^* \cap \mathbf{S}_a) \leq d$, we have:

$$\dim(\mathbf{W}) \geq k + (d - 1) - d = k - 1.$$

Now we analyze the properties of vectors in $\mathbf{W}$. Let $\mathbf{w} \in \mathbf{W}$. Since $\mathbf{w} \in \mathbf{S}_a$, we have $\mathbf{A}_t \mathbf{w} = 0$. Observe that

$$\mathbf{B}_t \mathbf{w} = (\mathbf{B}_{t-1} + \mathbf{A}_t^\top \mathbf{A}_t)\mathbf{w} = \mathbf{B}_{t-1}\mathbf{w} + \mathbf{A}_t^\top(\mathbf{A}_t \mathbf{w}) = \mathbf{B}_{t-1}\mathbf{w}.$$

Hence, $\mathbf{W}$ is a subspace contained in both $\mathbf{B}_{t-1}$ and $\mathbf{B}_t$.

Let $\lambda_1 \geq \ldots \geq \lambda_d$ be the eigenvalues of $\mathbf{B}_{t-1}$, and $\mu_1 \geq \ldots \geq \mu_d$ be the eigenvalues of $\mathbf{B}_t$. Since $\mathbf{B}_t$ is a rank-1 PSD update of $\mathbf{B}_{t-1}$, the eigenvalues interlace by Theorem A.3:

$$\mu_1 \geq \lambda_1 \geq \mu_2 \geq \lambda_2 \geq \ldots \geq \mu_k \geq \lambda_k \geq \mu_{k+1} \geq \lambda_{k+1}\ldots.$$

Since $\mathbf{W} \subseteq \mathbf{V}^* t - 1$, the subspace $\mathbf{W}$ is spanned by eigenvectors of $\mathbf{B}_{t-1}$ corresponding to eigenvalues $\lambda_1, \ldots, \lambda_k$. Because $\mathbf{B}_t \mathbf{w} = \mathbf{B}_{t-1}\mathbf{w}$ for $\mathbf{w} \in W$, these are also eigenvectors of $\mathbf{B}_t$ with the same eigenvalues. Since $\lambda_k \geq \mu_{k+1}$, the eigenvalues associated with the subspace $\mathbf{W}$ are greater than or equal to the $(k + 1)$-th eigenvalue of $\mathbf{B}_t$. Thus, $\mathbf{W}$ must be a subspace of the top-$(k + 1)$ eigenspace of $\mathbf{B}_t$. Therefore, $\mathbf{W} \subseteq \mathbf{V}_{t-1}^* \cap (\mathbf{V}_t^* \cup \{\mathbf{u}\})$, where $\mathbf{u}$ is the eigenvector of $\mathbf{B}_t$ corresponding to eigenvalue $\mu_{k+1}$. Since $\dim(\mathbf{W}) \geq k - 1$, we have established that $\dim(\mathbf{V}_{t-1}^* \cap \mathbf{V}_t^*) \geq k - 2$.

Now, let $\mathbf{P}_{t-1}$ and $\mathbf{P}_t$ be the orthogonal projection matrices onto $\mathbf{V}_{t-1}^*$ and $\mathbf{V}_t^*$, respectively. Let $\mathbf{W}_{\text{int}} = \mathbf{V}_{t-1}^* \cap \mathbf{V}_t^*$ and $\mathbf{P}_{\text{shared}}$ be the projection onto $\mathbf{W}_{\text{int}}$. If $\dim(\mathbf{W}_{\text{int}}) = k$, then $\mathbf{V}^*t - 1 = \mathbf{V}_t^*$ and so the recourse is $\|\mathbf{P}_t - \mathbf{P}_{t-1}\|_F^2 = 0$.

Otherwise, if $\dim(\mathbf{W}_{int}) = 1$, we can decompose the orthogonal projection matrices as:

$$\mathbf{P}_{t-1} = \mathbf{P}_{\text{shared}} + \mathbf{u}_1 \mathbf{u}_1^\top, \qquad \mathbf{P}^* t = \mathbf{P}_{\text{shared}} + \mathbf{u}_2 \mathbf{u}_2^\top,$$

where $\mathbf{u}_1, \mathbf{u}_2$ are unit vectors orthogonal to $\mathbf{W}_{\text{int}}$. The recourse is $\|\mathbf{P}_t - \mathbf{P}_{t-1}\|_F^2$. Thus, we have

$$\mathbf{P}_t - \mathbf{P}_{t-1} = (\mathbf{P}_{\text{shared}} + \mathbf{u}_2 \mathbf{u}_2^\top) - (\mathbf{P}_{\text{shared}} + \mathbf{u}_1 \mathbf{u}_1^\top) = \mathbf{u}_2 \mathbf{u}_2^\top - \mathbf{u}_1 \mathbf{u}_1^\top,$$

so that by generalized triangle inequality,

$$\text{Recourse}(\mathbf{P}_t, \mathbf{P}_{t-1}) \leq 2\|\mathbf{u}_1 \mathbf{u}_1^\top\|_F^2 + 2\|\mathbf{u}_2 \mathbf{u}_2^\top\|_F^2.$$

Since $\mathbf{u}_1$ and $\mathbf{u}_2$ are unit vectors, then we have $\text{Recourse}(\mathbf{P}_t, \mathbf{P}_{t-1}) \leq 4$. The same proof with four unit vectors shows that if $\dim(\mathbf{W}_{int}) = 2$, then $\text{Recourse}(\mathbf{P}_t, \mathbf{P}_{t-1}) \leq 8$. □

Finally, we remark that due to the simplicity of rank-one perturbations, any algorithm that maintains the SVD only needs to perform a rank-one update to the SVD, which takes time $\mathcal{O}\left((n+d)k + k^3\right)$ for a matrix with dimensions at most $n \times d$ Brand (2006).

# F  MISSING PROOFS FROM SECTION 3

**Lemma 3.3.** *Consider a time $s$ during which $c$ is reset to $0$. Suppose* HEAVY *is set to* FALSE *at time $s$ and $c$ is not reset to $0$ within the next $r$ steps, for $r \leq \sqrt{k}$. Let $\mathbf{V}^{(t)}$ be the output of $\mathbf{V}$ at time $t$. Then $\mathbf{V}^{(t)}$ provides a $\left(1 + \frac{\varepsilon}{2}\right)$-approximation to the cost of the optimal low-rank approximation of $\mathbf{A}^{(t)}$ for all $t \in [s, s + r]$.*

*Proof.* Consider $\|\mathbf{A}^{(t)} - \mathbf{A}^{(t)}(\mathbf{V}^{(t)})^\top \mathbf{V}^{(t)}\|_F^2$. Let $\mathbf{V}^{(t')}$ be the matrix $\mathbf{V}^{(s)}$ with the $t - s$ vectors corresponding to the smallest $t - s$ singular values of $\mathbf{A}^{(s)}$ instead being replaced with the rows $\mathbf{a}_{s+1}, \ldots, \mathbf{a}_t$. By optimality of $\mathbf{v}$, we have

$$\|\mathbf{A}^{(t)} - \mathbf{A}^{(t)}(\mathbf{V}^{(t)})^\top \mathbf{V}^{(t)}\|_F^2 \leq \|\mathbf{A}^{(t)} - \mathbf{A}^{(t)}(\mathbf{V}^{(t')})^\top \mathbf{V}^{(t')}\|_F^2.$$

Since $\mathbf{a}_t \in \mathbf{V}^{(t')}$ for all $t \in [s, s + \sqrt{k}]$, then we have

$$\|\mathbf{A}^{(t)} - \mathbf{A}^{(t)}(\mathbf{V}^{(t')})^\top \mathbf{V}^{(t')}\|_F^2 = \|\mathbf{A}^{(s)} - \mathbf{A}^{(s)}(\mathbf{V}^{(t')})^\top \mathbf{V}^{(t')}\|_F^2.$$

In other words, the rows $\mathbf{a}_{s+1}, \ldots, \mathbf{a}_t$ cannot contribute to the low-rank approximation cost of $\mathbf{V}^{(t')}$ because they are contained within the span of $\mathbf{V}^{(t')}$. It remains to upper bound $\|\mathbf{A}^{(s)} - \mathbf{A}^{(s)}(\mathbf{V}^{(t')})^\top \mathbf{V}^{(t')}\|_F^2$. We have

$$C = \sum_{i=k+1}^{d} \sigma_i^2(\mathbf{A}_s) = \|\mathbf{A}^{(s)} - \mathbf{A}^{(s)}(\mathbf{V}^{(s)})^\top \mathbf{V}^{(s)}\|_F^2$$

and $\sum_{i=k-\sqrt{k}}^{k} \sigma_i^2(\mathbf{A}_s) < \frac{\varepsilon}{3} \cdot C$ since HEAVY is set to FALSE. Note that since $t \in [s, s + \sqrt{k}]$, then $\mathbf{A}^{(t')}$ contains the top $k - \sqrt{k}$ singular vectors of $\mathbf{A}^{(s)}$. Therefore, it follows that

$$\|\mathbf{A}^{(s)} - \mathbf{A}^{(s)}(\mathbf{V}^{(t')})^\top \mathbf{V}^{(t')}\|_F^2 \leq \|\mathbf{A}^{(s)} - \mathbf{A}^{(s)}(\mathbf{V}^{(s)})^\top \mathbf{V}^{(s)}\|_F^2 + \sum_{i=k-\sqrt{k}}^{k} \sigma_i^2(\mathbf{A}_s) \leq \left(1 + \frac{\varepsilon}{3}\right) \cdot C.$$

Since the cost of the optimal low-rank approximation of $\mathbf{A}^{(t)}$ is at least the cost of the optimal low-rank approximation of $\mathbf{A}^{(s)}$ for $t > s$, then $\mathbf{V}^{(t)}$ provides a $\left(1 + \frac{\varepsilon}{3}\right)$-approximation to the cost of the optimal low-rank approximation of $\mathbf{A}^{(t)}$ for all $t \in [s, s + \sqrt{k}]$. □

**Lemma 3.6.** *Suppose* HEAVY *is set to* FALSE *at time $s$ and $c$ is reset to $0$ at time $s$. If $c$ is not reset to $0$ within the next $r$ steps, for $r \leq \sqrt{k}$, then $\sum_{i=s+1}^{s+r} \text{Recourse}(\mathbf{V}^{(s)}, \mathbf{V}^{(s-1)}) \leq r$.*

*Proof.* Let $s$ be a time during which $c$ is reset to $0$ and HEAVY is set to FALSE. Then for the next $r$ steps, each time a new row is received, then Algorithm 2 replaces a row of $\mathbf{V}$ with the new row. Thus, the recourse is at most $r$. □

**Lemma 3.4.** *Consider a time $t$ during which* HEAVY *is set to* TRUE. *Let $\mathbf{V}^{(t)}$ be the output of $\mathbf{V}$ at time $t$. Then $\mathbf{V}^{(t)}$ provides a $\left(1 + \frac{\varepsilon}{2}\right)$-approximation to the cost of the optimal low-rank approximation of $\mathbf{A}^{(t)}$.*

*Proof.* Let $\mathsf{OPT} = \sum_{i=k+1}^{d} \sigma^2(\mathbf{A}_t)$. We have two cases. Either $\|\mathbf{A}^{(t)} - \mathbf{A}^{(t)}(\mathbf{V}^{(t-1)})^\top \mathbf{V}^{(t-1)}\|_F^2 \geq \left(1 + \frac{\varepsilon}{2}\right) \cdot \mathsf{OPT}$ or $\|\mathbf{A}^{(t)} - \mathbf{A}^{(t)}(\mathbf{V}^{(t-1)})^\top \mathbf{V}^{(t-1)}\|_F^2 < \left(1 + \frac{\varepsilon}{2}\right) \cdot \mathsf{OPT}$.

In the former case, $\mathbf{V}^{(t-1)}$ is already a $\left(1 + \frac{\varepsilon}{2}\right)$-approximation to the cost of the optimal low-rank approximation of $\mathbf{A}^{(t)}$ and the algorithm sets $\mathbf{V}^{(t)} = \mathbf{V}^{(t-1)}$, so that $\mathbf{V}^{(t)}$ is also a $\left(1 + \frac{\varepsilon}{2}\right)$-approximation to the cost of the optimal low-rank approximation of $\mathbf{A}^{(t)}$.

In the latter case, the algorithm sets $\mathbf{V}^{(t)}$ to be the output of $\text{RECLUSTER}(\mathbf{A}^{(t)}, k)$, i.e., the top $k$ eigenvectors of $\mathbf{A}^{(t)}$, in which case $\|\mathbf{A}^{(t)} - \mathbf{A}^{(t)}(\mathbf{V}^{(t)})^\top \mathbf{V}^{(t)}\|_F^2 = \mathsf{OPT}$. Thus in both cases, $\mathbf{V}^{(t)}$ provides a $\left(1 + \frac{\varepsilon}{2}\right)$-approximation to the cost of the optimal low-rank approximation of $\mathbf{A}^{(t)}$. □

**Lemma 3.5.** *At all times $t \in [n]$, Algorithm 2 provides a $\left(1 + \frac{\varepsilon}{2}\right)$-approximation to the cost of the optimal low-rank approximation of $\mathbf{A}^{(t)}$.*

*Proof.* Let $\mathbf{V}^{(t)}$ be the output of $\mathbf{V}$ at time $t$. We first consider the times $t$ where $c$ is not reset to zero and HEAVY is set to FALSE. By Lemma 3.3, the output $\mathbf{V}^{(t)}$ is provides a $\left(1 + \frac{\varepsilon}{2}\right)$-approximation to the cost of the optimal low-rank approximation of $\mathbf{A}^{(t)}$ at these times.

We next consider the times $t$ where $c$ is not reset to zero and HEAVY is set to TRUE. By Lemma 3.4, the output $\mathbf{V}^{(t)}$ is provides a $\left(1 + \frac{\varepsilon}{2}\right)$-approximation to the cost of the optimal low-rank approximation of $\mathbf{A}^{(t)}$ at these times.

Finally, we consider the times $t$ where $c$ is reset to zero. At these times, the algorithm sets $\mathbf{V}^{(t)}$ to be the output of $\text{RECLUSTER}(\mathbf{A}^{(t)}, k)$, i.e., the top $k$ eigenvectors of $\mathbf{A}$, in which case $\|\mathbf{A}^{(t)} - \mathbf{A}^{(t)}(\mathbf{V}^{(t)})^\top \mathbf{V}^{(t)}\|_F^2 = \mathsf{OPT}$. Therefore, Algorithm 2 provides a $\left(1 + \frac{\varepsilon}{2}\right)$-approximation to the cost of the optimal low-rank approximation of $\mathbf{A}^{(t)}$ at all times $t \in [n]$. □

We next bound the recourse between epochs when the bottom $\sqrt{k}$ singular values of the top $k$ do not contribute significant mass.

**Lemma F.1.** *Consider a time $t$ during which* HEAVY *is set to* TRUE. *Let $\mathbf{V}^{(t)}$ be the output of $\mathbf{V}$ at time $t$. Suppose $\mathbf{V}^{(t-1)}$ fails to be a $\left(1 + \frac{\varepsilon}{2}\right)$-approximation to the optimal low-rank approximation cost. Then* $\text{Recourse}(\mathbf{V}^{(t)}, \mathbf{V}^{(t-1)}) \leq \sqrt{k}$.

*Proof.* Let $s$ be the most recent time at which HEAVY was set to TRUE, so that $s \leq t - 1$. Let $\mathbf{V}^{(s)}$ denote the top $k$ right singular vectors of $\mathbf{A}^{(s)}$ and note that by condition of HEAVY being set to TRUE,

$$\sum_{i=k-\sqrt{k}}^{k} \sigma_i^2(\mathbf{V}^{(s)}) \geq \frac{\varepsilon}{3} \cdot \|\mathbf{A}^{(s)} - \mathbf{A}^{(s)}(\mathbf{V}^{(s)})^\top \mathbf{V}^{(s)}\|_F^2.$$

Now, let $r$ be the time at which $\mathbf{V}^{(t-1)}$ was first set, so that $\mathbf{V}^{(t-1)}$ are the top $k$ right singular vectors of $\mathbf{A}^{(r)}$. Since $r \geq s$, then by the interlacing of singular values, i.e., Theorem A.3, we have that

$$\sigma_i(\mathbf{V}^{(r)}) \geq \sigma_i(\mathbf{V}^{(s)}),$$

for all $i \in [d]$. Therefore,

$$\sum_{i=k-\sqrt{k}}^{k} \sigma_i^2(\mathbf{V}^{(r)}) \geq \frac{\varepsilon}{3} \cdot \|\mathbf{A}^{(s)} - \mathbf{A}^{(s)}(\mathbf{V}^{(s)})^\top \mathbf{V}^{(s)}\|_F^2,$$

since all singular values are by definition non-negative.

Suppose by way of contradiction that we have

$$\text{Recourse}(\mathbf{V}^{(t)}, \mathbf{V}^{(t-1)}) = \text{Recourse}(\mathbf{V}^{(t)}, \mathbf{V}^{(r)}) > \sqrt{k}.$$

Then at least $\sqrt{k}$ singular vectors for $\mathbf{V}^{(t-1)}$ have been displaced and thus by the min-max theorem, i.e., Theorem A.2,

$$\|\mathbf{A}^{(t)} - \mathbf{A}^{(t)}(\mathbf{V}^{(t)})^\top \mathbf{V}^{(t)}\|_F^2 > \|\mathbf{A}^{(r)} - \mathbf{A}^{(s)}(\mathbf{V}^{(s)})^\top \mathbf{V}^{(s)}\|_F^2 + \sum_{i=k-\sqrt{k}}^{k} \sigma_i^2(\mathbf{V}^{(r)})$$

$$\geq \left(1 + \frac{\varepsilon}{3}\right) \cdot \|\mathbf{A}^{(s)} - \mathbf{A}^{(s)}(\mathbf{V}^{(s)})^\top \mathbf{V}^{(s)}\|_F^2.$$

Furthermore, because $\mathbf{V}^{(t)}$ is the top $k$ right singular vectors of $\mathbf{A}^{(t)}$, then $\|\mathbf{A}^{(t)} - \mathbf{A}^{(t)}(\mathbf{V}^{(t)})^\top \mathbf{V}^{(t)}\|_F^2$ is the cost of the optimal low-rank approximation at time $t$. In other words, the optimal low-rank approximation cost at time $t$ would be larger than $\left(1 + \frac{\varepsilon}{3}\right) \cdot \|\mathbf{A}^{(s)} - \mathbf{A}^{(s)}(\mathbf{V}^{(s)})^\top \mathbf{V}^{(s)}\|_F^2$.

On the other hand, since $s$ and $t$ are in the same epoch, then

$$\|\mathbf{A}^{(s)} - \mathbf{A}^{(s)}(\mathbf{V}^{(s)})^\top \mathbf{V}^{(s)}\|_F^2 \leq \|\mathbf{A}^{(t)} - \mathbf{A}^{(t)}(\mathbf{V}^{(t)})^\top \mathbf{V}^{(t)}\|_F^2 \leq \left(1 + \frac{\varepsilon}{4}\right) \|\mathbf{A}^{(s)} - \mathbf{A}^{(s)}(\mathbf{V}^{(s)})^\top \mathbf{V}^{(r)}\|_F^2,$$

which is a contradiction. Hence, it follows that $\text{Recourse}(\mathbf{V}^{(t)}, \mathbf{V}^{(t-1)}) \leq \sqrt{k}$. $\qquad\square$

**Lemma F.2.** *Suppose $\mathbf{A} \in \mathbb{Z}^{n \times d}$ is an integer matrix with rank $r > k$ and entries bounded in magnitude by $M$. Then the cost of the optimal low-rank approximation to $\mathbf{A}$ is at least $(ndM^2)^{-\frac{k}{r-k}}$.*

*Proof.* Let $\mathbf{A} \in \mathbb{Z}^{n \times d}$ be an integer matrix with rank $r > k$ and entries bounded in magnitude by $M$. Suppose without loss of generality that $n \geq d$. Let $\sigma_1 \geq \ldots \geq \sigma_d \geq 0$ be the singular values of $\mathbf{A}$ and let $\lambda_1 \geq \ldots \geq \lambda_d \geq 0$ be the corresponding eigenvalues of $\mathbf{A}^\top \mathbf{A}$. Let $p(\lambda) = \lambda^{d-r} \prod_{i \in [r]} (\lambda - \lambda_i)$. Note that since the entries of $\mathbf{A}$ are all integers, then the entries of $\mathbf{A}^\top \mathbf{A}$ are all integers and thus the coefficients of $p(\lambda)$ are all integers. In particular, since the coefficient of $\lambda^{d-r}$ in $p(\lambda)$ is the product of the nonzero eigenvalues of $\mathbf{A}^\top \mathbf{A}$, then $\prod_{i=1}^{r} \lambda_i \geq 1$. Since $\lambda_i = \sigma_i^2$ and $\sigma_i \geq 0$ for all $i \in [d]$, we also have $\prod_{i=1}^{r} \sigma_i \geq 1$.

Moreover, the squared Frobenius norm satisfies

$$\sum_{i=1}^{d} \lambda_i = \sum_{i=1}^{d} \sigma_i^2 = \|\mathbf{A}\|_F^2 \leq ndM^2.$$

Thus, $\lambda_i \leq ndM^2$ for all $i \in [d]$. Hence,

$$\lambda_{k+1}^{r-k} \geq \prod_{i=k+1}^{r} \lambda_i \geq \frac{1}{(ndM^2)^k} \prod_{i=1}^{r} \lambda_i \geq \frac{1}{(ndM^2)^k}.$$

Thus we have $\lambda_{k+1} \geq (ndM^2)^{-\frac{k}{r-k}}$. It follows that the optimal low-rank approximation cost is

$$\sqrt{\sum_{i=k+1}^{d} \lambda_i} \geq \lambda_{k+1} \geq (ndM^2)^{-\frac{k}{r-k}}.$$

$\qquad\square$

**Lemma 3.7.** *Suppose HEAVY is set to TRUE at time $t$ and $c$ is not reset to $0$ within the next $r$ steps, for $r \leq k$. Then $\sum_{i=t+1}^{t+r} \text{Recourse}(\mathbf{V}^{(t)}, \mathbf{V}^{(t-1)}) \leq r\sqrt{k}$.*

*Proof.* By Lemma F.1, we have $\text{Recourse}(\mathbf{V}^{(t)}, \mathbf{V}^{(t-1)}) \leq \sqrt{k}$ for all $i \in [t+1, t+r]$. Thus the total recourse is at most $r\sqrt{k}$. $\qquad\square$

We analyze the total recourse during times when we reset the counter $c$ because the cost of the optimal low-rank approximation has doubled.

**Lemma F.3.** *Let $T$ be the set of times at which $c$ is set to $0$ because $\sum_{i=k+1}^{d} \sigma_i^2(\mathbf{A}_t) \geq 2C$. Then $\sum_{t \in T} \text{Recourse}(\mathbf{V}^{(t)}, \mathbf{V}^{(t-1)}) \leq \mathcal{O}\left(\frac{k}{\varepsilon} \log^2(ndM)\right)$.*

*Proof.* We define times $\tau_1 < \tau_2$ and decompose $T$ into the times before $\tau_1$, the times between $\tau_1$ and $\tau_2$, and the times after $\tau_2$. Formally, let $t_0$ be the first time at which the optimal low-rank approximation cost is nonzero, i.e., the first time at which the input matrix has rank $k + 1$. Let $T_0$ be the optimal low-rank approximation cost at time $t_0$. Define each epoch $i$ to be the times during which the cost of the optimal low-rank approximation is at least $\left(1 + \frac{\varepsilon}{4}\right)^i \cdot T_0$ and less than $\left(1 + \frac{\varepsilon}{4}\right)^{i+1} \cdot T_0$.

Let $\tau_1$ be the first time the input matrix has rank $k + 1$. Observe that before time $\tau_1$, we can maintain the entire row span of the matrix by adding each new linearly independent row to the low-rank subspace, thus preserving the optimal low-rank approximation cost at all times. This process incurs recourse at most $k$ in total before time $\tau_1$.

Next, let $\tau_2$ be the first time such that the input matrix has rank at most $2k$. We analyze the recourse between times $\tau_1$ and $\tau_2$. By Lemma F.2, the cost of the optimal low-rank approximation to a matrix with integer entries bounded by $M$ and rank $r$ is at least $(ndM^2)^{-\frac{k}{r-k}}$. Thus if the rank $r$ of the matrix is at least $k + 2^i$ and less than $k + 2^{i+1}$, then the cost of the optimal low-rank approximation is at least $(ndM^2)^{-\frac{k}{2^i}}$. Thus there can be at most $\mathcal{O}\left(\frac{1}{\varepsilon}\frac{k}{2^i}\right)$ epochs before the cost of the optimal low-rank approximation is at least $(ndM^2)^{-100}$. Let $j$ be the index of any epoch during which the cost of the optimal low-rank approximation exceeds $(ndM^2)^{-\frac{k}{2^i}}$. Since the total dimension of the span of the rows of $\mathbf{A}$ that have arrived by epoch $j$ is $r$, then the recourse incurred by recomputing the top eigenspace is at most $\mathcal{O}(r - k) = \mathcal{O}(2^i)$. Since there can be at most $\mathcal{O}\left(\frac{1}{\varepsilon}\frac{k}{2^i}\log(ndM)\right)$ epochs before the cost of the optimal low-rank approximation is at least $(ndM^2)^{-100}$, then the consistency cost for the times where the rank of the matrix is at least $k + 2^i$ and less than $k + 2^{i+1}$ is at most $\mathcal{O}\left(\frac{k}{\varepsilon}\log(ndM)\right)$. Thus the total consistency cost between times $\tau_1$ and $\tau_2$ is $\sum_{i=0}^{\log k} \mathcal{O}\left(\frac{k}{\varepsilon}\log(ndM)\right) = \mathcal{O}\left(\frac{k}{\varepsilon}\log^2(ndM)\right)$.

Note that after time $\tau_2$, the cost of the optimal low-rank approximation is at least $(ndM^2)^{-100}$. Since the squared Frobenius norm is at most $ndM^2$, then the low-rank cost is also at most $ndM^2$. Thus there can be at most $\mathcal{O}\left(\frac{1}{\varepsilon}\log(ndM)\right)$ epochs after time $\tau_2$. Each epoch incurs recourse $\mathcal{O}(k)$ due to recomputing the top eigenspace of the prefix of $\mathbf{A}$ that has arrived at that time. Thus the total recourse due to the set $T$ of times after $\tau_2$ is $\mathcal{O}\left(\frac{k}{\varepsilon}\log(ndM)\right)$.

In summary, we can decompose $T$ into the times before $\tau_1$, the times between $\tau_1$ and $\tau_2$, and the times after $\tau_2$. The total recourse at times $t \in T$ before $\tau_1$ is at most $\mathcal{O}(k)$. The total recourse at times $t \in T$ between times $\tau_1$ and $\tau_2$ is at most $\mathcal{O}\left(\frac{k}{\varepsilon}\log^2(ndM)\right)$. The total recourse at times $t \in T$ after time $\tau_2$ is at most $\mathcal{O}(k\log(ndM))$. Hence, the total recourse of times $t \in T$ is at most $\mathcal{O}\left(\frac{k}{\varepsilon}\log^2(ndM)\right)$. $\square$

**Lemma 3.8.** *Let $T$ be the set of times at which $c$ is set to $0$. Then $\sum_{t \in T} \text{Recourse}(\mathbf{V}^{(t)}, \mathbf{V}^{(t-1)}) \leq \mathcal{O}\left(n\sqrt{k} + \frac{k}{\varepsilon}\log^2(ndM)\right)$.*

*Proof.* Note that $c$ can only be reset for one of the three different following reasons:

(1) HEAVY = TRUE and $c = k$

(2) HEAVY = FALSE and $c = \sqrt{k}$

(3) $\sum_{i=k+1}^{d} \sigma_i^2(\mathbf{A}_t) \geq 2C$

In all three cases, the algorithm calls $\text{RECLUSTER}(\mathbf{A}, k)$, incurring recourse $k$. Observe that for a matrix $\mathbf{A}$ with $n$ rows, the counter $c$ can exceed $\sqrt{k}$ at most $\frac{n}{\sqrt{k}}$ times. Thus the first two cases can occur at most $\frac{n}{\sqrt{k}}$ times, so the total recourse contributed by the first two cases is at most $n\sqrt{k}$.

It remains to consider the total recourse incurred over the steps where the cost of the optimal low-rank approximation has at least doubled from the previous time $C$ was set. By Lemma F.3, the recourse from such times is at most $\mathcal{O}\left(\frac{k}{\varepsilon}\log^2(ndM)\right)$. Hence, the total recourse is at most $\mathcal{O}\left(n\sqrt{k} + \frac{k}{\varepsilon}\log^2(ndM)\right)$. $\qquad\square$

**Lemma 3.9.** *The total recourse of Algorithm 2 on an input matrix $\mathbf{A} \in \mathbb{R}^{n \times d}$ with integer entries bounded in magnitude by $M$ is $\mathcal{O}\left(n\sqrt{k} + \frac{k}{\varepsilon}\log^2(ndM)\right)$.*

*Proof.* We first consider the times $t$ where $c$ is not reset to zero and HEAVY is set to FALSE By Lemma 3.6, the recourse across any consecutive set of $r$ of these steps is at most $r$. Thus over the stream of $n$ rows, the total recourse incurred across all steps where HEAVY is set to FALSE is at most $n$.

We next consider the times $t$ where $c$ is not reset to zero and HEAVY is set to TRUE. By Lemma 3.7, the recourse across any uninterrupted sequence of $r$ these steps is at most $r\sqrt{k}$. Hence over the stream of $n$ rows, the total recourse incurred across all steps where HEAVY is set to TRUE is at most $n\sqrt{k}$.

Finally, we consider the times $t$ where $c$ is reset to zero. By Lemma 3.8, the total recourse incurred across all steps is at most $\mathcal{O}\left(n\sqrt{k} + \frac{k}{\varepsilon}\log^2(ndM)\right)$. Therefore, the total recourse is $\mathcal{O}\left(n\sqrt{k} + \frac{k}{\varepsilon}\log^2(ndM)\right)$. $\qquad\square$

**Lemma 3.10.** *Given an input matrix $\mathbf{A} \in \mathbb{R}^{n \times d}$ with integer entries bounded in magnitude by $M$, Algorithm 2 achieves a $\left(1 + \frac{\varepsilon}{2}\right)$-approximation to the cost of the optimal low-rank approximation and achieves recourse $\mathcal{O}\left(n\sqrt{k} + \frac{k}{\varepsilon}\log^2(ndM)\right)$.*

*Proof.* Note that correctness follows from Lemma 3.5 and the upper bound on recourse follows from Lemma 3.9. $\qquad\square$

**Theorem 1.3.** *Suppose $\mathbf{A} \in \mathbb{Z}^{n \times d}$ is an integer matrix with entries bounded in magnitude by $M$. There exists an algorithm that achieves a $(1 + \varepsilon)$-approximation to the cost of the optimal low-rank approximation $\mathbf{A}$ at all times and achieves recourse $\frac{k^{3/2}}{\varepsilon^2} \cdot \text{polylog}(ndM)$.*

*Proof.* Let $\varepsilon \in \left(0, \frac{1}{100}\right)$. By Theorem 2.4, the optimal low-rank approximation to the rows of $\mathbf{M}$ that have been sampled by a time $t$ achieves a $\left(1 + \frac{\varepsilon}{10}\right)$-approximation to the cost of the optimal low-rank approximation of $\mathbf{A}$ that have arrived at time $t$. Thus, it suffices to show that we provide a $\left(1 + \frac{\varepsilon}{2}\right)$-approximation to the cost of the optimal low-rank approximation to the matrix $\mathbf{M}$ at all times. Thus we instead consider a new stream consisting of the rows of $\mathbf{M} \in \mathbb{R}^{m \times d}$, where $m = \frac{k}{\varepsilon^2} \cdot \text{polylog}(ndM)$ and the entries of $\mathbf{M}$ are integers bounded in magnitude by $M \cdot \text{poly}(n)$.

Consider Algorithm 2 on input $\mathbf{M}$. Correctness follows from Lemma 3.10, so it remains to reparameterize the settings in Lemma 3.10 to analyze the total recourse. By Lemma 3.9, the total recourse on an input matrix $\mathbf{A} \in \mathbb{R}^{n \times d}$ with integer entries bounded in magnitude by $M$ is $\mathcal{O}\left(n\sqrt{k} + \frac{k}{\varepsilon}\log^2(ndM)\right)$. Thus for input matrix $\mathbf{M}$ with $\frac{k}{\varepsilon^2} \cdot \text{polylog}(ndM)$ rows and integer entries bounded in magnitude by $M \cdot \text{poly}(n)$, the total recourse is $\frac{k^{3/2}}{\varepsilon^2} \cdot \text{polylog}(ndM)$. $\qquad\square$

## G  ADDITIONAL EXPERIMENTS

In this section, we describe a number of additional empirical evaluations.

### G.1 RANDOM SYNTHETIC DATASET

We generate a random synthetic dataset with 3000 rows and 4 columns with integer entries between 0 and 100 and subsequently normalized by column. We again compared the cost of the solution output by Algorithm 4 with the cost of the optimal low-rank approximation for $k = 1$ across $c = (1 + \varepsilon) \in \{1.1, 2.5, 5, 10, 100\}$. We summarize our results in Figure 3. In particular, we first plot in Figure 3a the runtime of our algorithm. We then plot in Figure 3b the ratio of the cost of the solution output by Algorithm 4 with the cost of the optimal low-rank approximation.

Similar to the skin segmentation and the rice datasets, our algorithm provides better approximation to the optimal solution as $c = (1 + \varepsilon)$ decreases from 100 to 1, with a number of spikes for a small number of rows likely due to the optimal low-rank approximation cost being quite small compared to the additive Frobenius error. Moreover, our results perform demonstrably better than the worst-case theoretical guarantee, giving roughly a 2.5-approximation compared to the theoretical guarantee of 100-approximation.

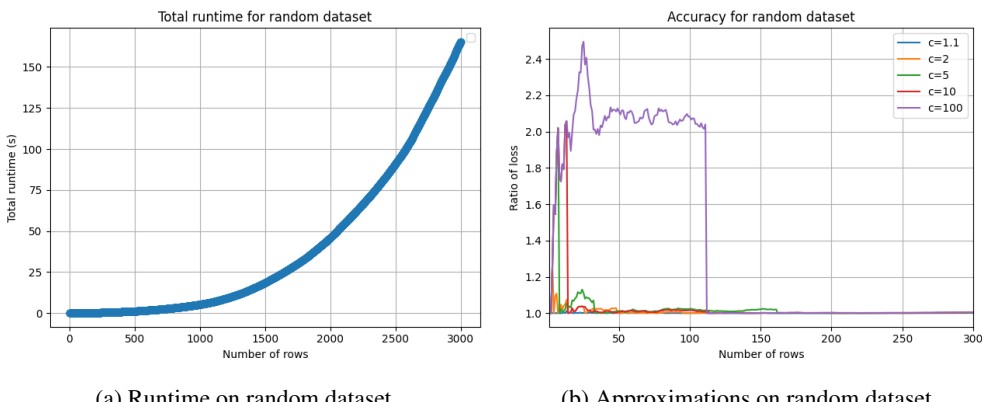

(a) Runtime on random dataset        (b) Approximations on random dataset

Fig. 3: Runtime and approximations on random dataset. Figure 3a considers $k = 1$, $c = 10$, while Figure 3b considers $k = 1$, $c = (1 + \varepsilon) \in \{1.1, 2.5, 5, 10, 100\}$

### G.2 SKIN SEGMENTATION DATASET

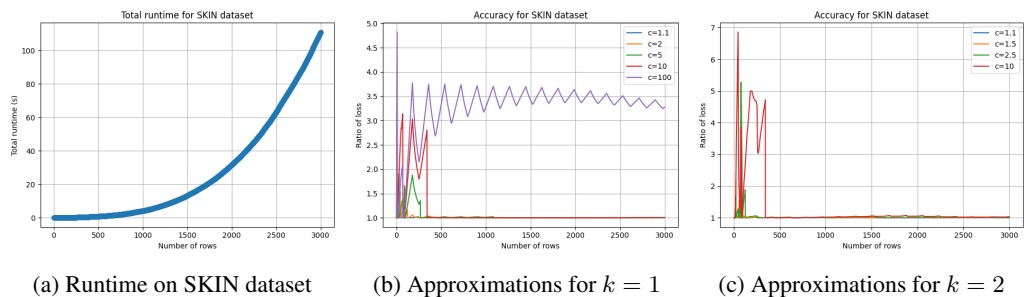

(a) Runtime on SKIN dataset    (b) Approximations for $k = 1$    (c) Approximations for $k = 2$

Fig. 4: Runtime and approximations on SKIN dataset. Figure 4a considers $k = 1$ and $c = 1.1$, while Figure 4b considers $k = 1$, $c = (1 + \varepsilon) \in \{1.1, 2.5, 5, 10, 100\}$ and Figure 4c considers $k = 2$, $c = (1 + \varepsilon) \in \{1.1, 1.5, 2.5, 10\}$

We evaluate Algorithm 4 on the Skin Segmentation (SKIN) dataset (Bhatt & Dhall, 2012) from the UCI repository (Markelle Kelly, 1987), which is commonly used in benchmark comparison for unsupervised learning tasks, e.g., (Borassi et al., 2020; Epasto et al., 2023; Woodruff et al., 2023). The dataset consists of a total of 245057 face images encoded by B,G,R values, collected from the Color FERET Image Database and the PAL Face Database from Productive Aging Laboratory from the University of Texas at Dallas. The faces were collected from various age groups (young, middle, and old), race groups (white, black, and asian), and genders. Among the dataset, 50859 images are

skin samples, while the other 194198 images are non-skin samples, and the task is to classify which category each image falls under.

**Experimental setup.** For our experiments, we only considered the first 3000 skin images and stripped the labels, so that the goal was to perform low-rank approximation on the B,G,R values of the remaining skin images. As our theoretical guarantees ensure that the solution is changed a small number of times, we compared the cost of the solution output by Algorithm 4 with the cost of the optimal low-rank approximation. In particular, we computed the ratios of the two costs for $k = 1$ across $c = (1 + \varepsilon) \in \{1.1, 2.5, 5, 10, 100\}$ and for $k = 2$ across $c = (1 + \varepsilon) \in \{1.1, 1.5, 2.5, 10\}$, even though the formal guarantees of Algorithm 4 involve upper bounding the additive error.

**Results and discussion.** In Figure 4, we plot the ratio of the cost of the solution output by Algorithm 4 with the cost of the optimal low-rank approximation at each time over the duration of the data stream. We then provide the central statistics, i.e., the mean, standard deviation, and maximum for the ratio of across various values of $k$ and accuracy parameters for the SKIN dataset in Table 2.

Our results show that as expected, our algorithm provides better approximation to the optimal solution as $c = (1 + \varepsilon)$ decreases from 100 to 1. Once the optimal low-rank approximation cost became sufficiently large, our algorithm achieved a good multiplicative approximation. Thus we believe the main explanation for the spikes at the beginning of Figure 4c is due to the optimal low-rank approximation cost being quite small compared to the additive Frobenius error. It is somewhat surprising that despite the worst-case theoretical guarantee that our algorithm should only provide a 100-approximation, it actually performs significantly better, i.e., it provides roughly a 4-approximation. Thus it seems our empirical evaluations provide a simple proof-of-concept demonstrating that our theoretical worst-case guarantees can be even stronger in practice.

| $k$ | $(1 + \varepsilon)$ | Mean | Std. Dev. | Max |
|---|---|---|---|---|
| | 1.1 | 1.0006 | 0.0022 | 1.0383 |
| | 2 | 1.0088 | 0.0479 | 1.7870 |
| 1 | 5 | 1.0374 | 0.1341 | 2.6038 |
| | 10 | 1.1324 | 0.4167 | 4.8201 |
| | 100 | 3.2971 | 0.4532 | 4.8201 |
| | 1.1 | 1.0016 | 0.0069 | 1.1521 |
| 2 | 1.5 | 1.0148 | 0.1327 | 5.1533 |
| | 2.5 | 1.0371 | 0.2430 | 5.2778 |
| | 10 | 1.3175 | 0.9238 | 6.8602 |

Table 2: Average, standard deviation, and maximum for ratios of cost across various values of $k$ and accuracy parameters for SKIN dataset.

### G.3 RICE DATASET

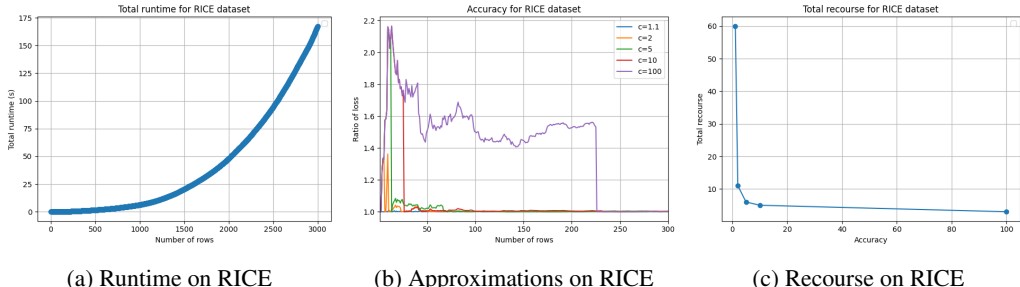

(a) Runtime on RICE  (b) Approximations on RICE  (c) Recourse on RICE

Fig. 5: Runtime and approximations on RICE dataset. Figure 5a considers $k = 1$, $c = 10$, while Figure 5b and Figure 5c consider $k = 1$, $c = (1 + \varepsilon) \in \{1.1, 2.5, 5, 10, 100\}$

We next consider the RICE dataset (Rice) from the UCI repository (Markelle Kelly, 1987), where the goal is to classify between 2 types of rice grown in Turkey. The first type of rice is the Osmancik species, which has a large planting area since 1997, while the second type of rice is the Cammeo

species, which has been grown since 2014 (Rice). The dataset consists of a total of 3810 rice grain images taken for the two species, with 7 morphological features were obtained for each grain of rice. Specifically, the features are the area, perimeter, major axis length, minor axis length, eccentricity, convex area, and the extent of the rice grain.

**Evaluation summary.** For our experiments, we performed low-rank approximation on the seven provided features of the RICE dataset. We compared the cost of the solution output by Algorithm 4 with the cost of the optimal low-rank approximation for $k = 1$ across $c = (1 + \varepsilon) \in \{1.1, 2.5, 5, 10, 100\}$. We summarize our results in Figure 5, plotting the runtime of our algorithm in Figure 5a and the ratio of the cost of the solution output by Algorithm 4 with the cost of the optimal low-rank approximation in Figure 5b.

Our results show that similar to the skin segmentation dataset, our algorithm provides better approximation to the optimal solution as $c = (1 + \varepsilon)$ decreases from 100 to 1. Figure 5b again has a number of spikes for a small number of rows likely due to the optimal low-rank approximation cost being quite small compared to the additive Frobenius error. Furthermore, our results again exhibit the somewhat surprising result that our algorithm provides a relatively good approximation compared to the worst-case theoretical guarantee, i.e., our algorithm empirically provides roughly a 2-approximation despite the worst-case guarantees only providing a 100-approximation.

Finally, we note that, as anticipated, the recourse of our algorithm decreases as the required accuracy of the factors decreases. This is because coarser factor representations require less frequent updates as the matrix evolves.

