# OpenReview forum: "Consistent Low-Rank Approximation"
_ICLR.cc/2026/Conference — ICLR 2026 Poster_

### Official Review · Reviewer_vaVm · 2025-10-25

**Soundness:** 4
**Presentation:** 4
**Contribution:** 3
**Rating:** 8
**Confidence:** 5

**Summary:**

This paper studies low rank approximation in a streaming model, where in addition to standard goals of small space and update time, they also do not want the provided solution to change too much across the lifetime of the stream.  This is modeled as "recourse" which is the sum of squared distances between subspaces at each step.

**Strengths:**

Standard streaming subspace approximation algorithms like FrequentDirections and Ridge Leverage Score Sampling can have very large recourse as shown theoretically, and empirically on real data.  That means they can bounce between solutions.

The algorithm is subtle yet simple.  It is careful about when to update the estimate with extra care to not to change the subspace too much if it does not have to.  It reminds me of distributed streaming algorithms (e.g., https://arxiv.org/abs/1404.7571) that try to minimize total communication of updates, but with focus on ensuring a stable answer.

I think the recourse setting is natural and useful.  It is a nice way to quantify stability of the sketch.

A strength is that feels like a complete paper on this topic.  It has a variety of upper bounds for additive and relative error, and shows lower bounds on recourse that asymptotically match the upper bounds.  There are basic experiments that show that the algorithm is not just theoretical, but works in practice -- whereas baselines like FrequentDirections does not.

**Weaknesses:**

Nothing to note.

**Questions:**

None.

---

> ### Author Response · Authors · 2025-11-21
>
> We are grateful for the very positive and detailed review. We appreciate the reviewer recognizing the subtlety of our algorithm, the naturalness of the recourse setting, and the completeness of our results. We are encouraged that the reviewer finds this work to be a strong contribution.

---

### Official Review · Reviewer_odAo · 2025-10-26

**Soundness:** 3
**Presentation:** 3
**Contribution:** 2
**Rating:** 8
**Confidence:** 2

**Summary:**

This paper studies the online low rank approximation problem. In this problem one is given a matrix $A\in \mathbb{R}^{n\times d}$ with integer entries bounded by $M$ and whose rows $a_1,\ldots, a_n$ arrive one by one. Let $A^{t}$ denote the matrix of the first $t$ rows at time $t\in [n]$, the goal is to output a matrix $V^{t}\in \mathbb{R}^{k\times n}$ such that $A^{t}(V^{t})^T V^{t}$ is a $1+\epsilon$ approximation rank $k$ approximation to $A^{t}$ at every time $t\in [n]$. In particular the paper studies \emph{consistent} algorithms for online low rank approximation. More precisely the goal is to minimize the recourse of the algorithm measured as

$$\sum_{t=1}^n \|P_A-P_B\|_F^2$$

for $A = V_t$ and $B = V_{t-1}$ where $P_{V}$ is the orthogonal projection matrix of the subspace spanned by $V$. Thus a low recourse algorithm ensures that the subspace of low rank approximation does not change drastically on average over the stream. Note that recourse of $nk$ can be achieved trivially by computing the best rank $k$ approximation at each step from scratch.

The first result shown in the paper is an algorithm that achieves a recourse of $O((k/\epsilon)\log(ndM))$ but incurs an additional additive error $\epsilon \|A^{t}\|_F^2$ at each step $t\in [n]$. Furthermore they show that a recourse of $O((k/\epsilon^2)\log^2 n)$ assuming an online condition number of poly$(n)$ and no additive error. Finally for matrices with integer entries they also obtain improved bounds. On the negative side they prove a lower bound on the recourse of $\Omega(n/\epsilon \log(n/k))$ for obtaining a $1+\epsilon$ approximation at every time step by constructing a hard sequence of rows.

**Strengths:**

The paper introduces a novel model for studying low rank approximation of consistency. Consistent and low recourse algorithms have been studied for other problems in data science thus making low rank approximation a natural problem to study from a theoretical perspective. Moreover the authors show good upper and lower bounds for low rank approximation in this model.

**Weaknesses:**

The paper does not have many weaknesses but one is that although the problem has a natural theoretical motivation, it would be interesting for the authors to discuss more concrete practical motivations for studying low recourse algorithms for low rank approximation.

**Questions:**

Although the authors very briefly discuss potential applications in feature engineering, it would be interesting to see if there are any concrete applications of low recourse algorithms for low rank approximation.

---

> ### Author Response · Authors · 2025-11-21
>
> > The paper does not have many weaknesses but one is that although the problem has a natural theoretical motivation, it would be interesting for the authors to discuss more concrete practical motivations for studying low recourse algorithms for low rank approximation.
>
> > Although the authors very briefly discuss potential applications in feature engineering, it would be interesting to see if there are any concrete applications of low recourse algorithms for low rank approximation.
>
> Thank you for this question. The primary motivation comes from machine learning pipelines where low-rank approximation (LRA) is used for feature engineering (L76–79). In these systems, the factors produced by LRA define the feature subspace, which is then used as input for downstream models, such as for prediction, recommendation, or representation learning. The quality and stability of these features directly affect the performance and reliability of the models that rely on them.
>
> If an online LRA algorithm has high recourse, even small updates to the data can cause large changes in the feature subspace. This instability forces frequent and costly retraining of downstream models to adapt to the new representations. Low-recourse algorithms address this issue by ensuring that the feature space evolves smoothly over time, reducing retraining costs and improving model consistency and reliability in production environments.
>
> Stable features are important across many applications. In biometrics, they ensure reliable identification; in image processing, they support object detection, handwriting recognition, and facial recognition; and in text mining and information retrieval, they maintain consistent embeddings or TF–IDF representations for search and classification. Even in large-scale data curation for foundation models, where clustering, a form of LRA, selects representative data, low-recourse algorithms prevent unstable changes and reduce repeated retraining. We have included an expanded version of this discussion in the updated manuscript, with more details about each of these applications, to emphasize the practical importance of consistency in real-world dynamic systems.

---

### Official Review · Reviewer_ex4U · 2025-10-31

**Soundness:** 3
**Presentation:** 2
**Contribution:** 2
**Rating:** 4
**Confidence:** 3

**Summary:**

This paper studies the problem of low-rank approximation (LRA). Specifically given a matrix $A$, this work studies the problem of approximating $A$ with a matrix $AV^TV$, such that $||A-AV^TV||_F^2 \leq (1+\epsilon)||A-A_k||_F^2$ where $A_k$ is the best rank-$k$ approximation of $A$, and rows of $A$ arrive sequentially in time. This is a very widely studied problem. A primary contribution of the paper is the following problem: given rows of $A$ arrive sequentially over time, define measure called recourse computed as $||P_t - P_t-1||_F^2$ where $P_t$ is the orthogonal projection matrix corresponding to $V^TV$ at time $t$. This work studies LRA through the lens of recourse and demonstrates that -- 1) when the goal is to approximate $A$ with $\epsilon$ additive error, an $O(k\log(nd)/\epsilon)$ recourse is feasible, 2) when the goal is to approximate $A$ with $1+\epsilon$ multiplicative error, an $O(k^2\text{poly}\log(nd)/\epsilon^2)$ recourse is feasible. This is further improved to $k^{3/2}\text{poly}\log(nd)/\epsilon^2$ for matrices with integer entries that are bounded, and $k^{2}\text{poly}\log(nd)/\epsilon^2$ when condition number is bounded. A lower bound of $\Omega(k\log(n/k)/\epsilon)$ is also shown for $1+\epsilon$ multiplicative approximation algorithms.

**Strengths:**

- The problem setting is interesting, i.e., studying of the subspace corresponding to streaming updates and understand how subspace can differ for different algorithms is an interesting idea. Mostly because one can imagine having to reconstruct the approximation matrix again and again if the subspace is changing significantly (e.g., as stated for the Frequent directions method).

- I have only glossed over the proofs, which are pretty simple, and believe they are correct. Given the authors present a lower bound to the problem, it helps us ground the theoretical upper bounds presented in this work.

- I really appreciate the simple algorithms which helps maintain the approximation at time $t$. The algorithm basically checks importance of an incoming row by first identifying the bottom $\sqrt{k}$ singular values among the top $k$ singular vectors. If these vectors have very low spectral contribution, they are "disposable" and so can be replaced by any incoming vector.

- Good empirical evaluations help us understand how the algorithms presented here work in practice.

**Weaknesses:**

- There is a significant body of work on rank-$k$ approximation algorithms. However, only frequent directions has been empirically compared against. I am surprised as why this is the case.

- Most of the theoretical contributions are really derivative of prior work. While I really appreciate the problem setting, the contributions are really understanding how the subspace are drifting with time given the subspace approximation algorithm.

- Algorithm 2 requires computing SVD at each round in the worst case. So while one may be easily able to reduce recourse, the run time of the algorithms grows with $tk^3$, which seems expensive!

- For distributional shifts, just checking the bottom $\sqrt{k}$ may not be enough, e.g., for windowed algorithms due to Braverman et. al. 2020, or the works of Musco-Musco, or Cohen et. al. on online leverage score sampling, one might need to re-evaluate samples which was heavy at some point and might become of low importance in future. What do we do then?

**Questions:**

Please check the weaknesses section.

---

> ### Author Response · Authors · 2025-11-21
>
> > There is a significant body of work on rank-$k$ approximation algorithms. However, only frequent directions has been empirically compared against. I am surprised as why this is the case.
>
> Our focus is specifically on consistency (low recourse). To the best of our knowledge, existing streaming algorithms for low-rank approximation are not designed for this objective. We selected Frequent Directions (FD) as a representative, state-of-the-art deterministic streaming algorithm. As we discuss in Section 1.2 (L175-189) and show empirically, standard methods like FD can incur linear recourse, failing catastrophically. This is because they typically recompute the SVD of a sketched matrix, which can oscillate between nearly orthogonal subspaces even with small input changes (L181-189). Since our algorithms are the first designed for consistent LRA, we believe FD serves as the most relevant standard baseline to highlight the motivation for studying this problem.
>
> > Most of the theoretical contributions are really derivative of prior work. While I really appreciate the problem setting, the contributions are really understanding how the subspace are drifting with time given the subspace approximation algorithm.
>
> We respectfully emphasize our novel technical contributions. While we use tools like online ridge leverage score sampling, one of our core technical innovations is **Algorithm 2**, which handles the case where the optimal low-rank approximation cost can be exponentially small. A naive application of online ridge leverage score sampling yields $k^{2}\cdot \mathrm{poly}\log(n)$ recourse (L140–147).
>
> We introduce novel casework (L297–304) based on the spectral mass of the bottom $k$ singular vectors to balance small updates vs. full recomputation, reducing recourse to $k^{3/2}\cdot \mathrm{poly}\log(n)$, even when the optimal low-rank approximation solution has cost that is exponentially small.
>
> In particular, handling anti-Hadamard matrices (L369–375) in this case required developing a new structural property (Lemma F.2) and a rank-dependent analysis (L733–751), which are crucial and novel contributions. Finally, when the optimal low-rank approximation cost is at least some (inverse) polynomial in $n$, we further improve these bounds, obtaining an algorithm with recourse $k\cdot \mathrm{poly}\log(n)$
>
> > Algorithm 2 requires computing SVD at each round in the worst case. So while one may be easily able to reduce recourse, the run time of the algorithms grows with $tk^3$, which seems expensive!
>
> We clarify that our algorithm does not compute SVD at every step. **Algorithm 3** first uses online RLS to reduce the effective stream length from $n$ to $m = O(k/\epsilon^{2} \cdot \mathrm{poly}\log(ndM))$. **Algorithm 2** then runs on this reduced stream and calls the subroutine RECLUSTER (i.e., SVD) only sparingly—when the cost increases significantly or after $k$ (or $k$) small updates (Alg 2, L4). Often, it performs only a rank-1 update (L15).
>
> Consequently, the amortized update time is efficient: $d \cdot \mathrm{poly}(k, 1/\epsilon, \log(ndM))$ (L153, L414), and does not grow linearly with $n$.
>
> > For distributional shifts, just checking the bottom $\sqrt{k}$ may not be enough, e.g., for windowed algorithms due to Braverman et. al. 2020, or the works of Musco-Musco, or Cohen et. al. on online leverage score sampling, one might need to re-evaluate samples which was heavy at some point and might become of low importance in future. What do we do then?
>
> This is an excellent question. Our work initiates the study of consistent LRA and focuses on the foundational insertion-only streaming model, following the precedent in consistent clustering (e.g., Lattanzi & Vassilvitskii, 2017). The sliding window model (handling deletions/distributional shifts) introduces significant challenges, as the implicit removal of important rows can legitimately cause large, necessary shifts in the optimal subspace. Adapting our techniques to handle these implicit shifts efficiently is a compelling direction for future research.
>
> Moreover, we remark that our Algorithm 1 can handle the case when the distribution shift is more explicit, i.e., previous rows are explicitly deleted such as in the insertion-deletion model, as opposed to implicit expirations such as in the sliding window model. Specifically, Lemma 2.1 can be restated to show that Algorithm 2 will achieve constant recourse per update, regardless of whether the update is an insertion or a deletion.

---

> > ### Comment · Reviewer_ex4U · 2025-11-28
> > **Response to the comments by the authors**
> >
> > I am a little bit confused when the authors say
> > > existing streaming algorithms for low-rank approximation are not designed for this objective.
> >
> > The whole body of work on subspace approximations for streaming algorithms are designed to approximate subspaces in various kinds of streaming setting. So to say that FD is the only SoTA algorithm out there is confusing to me.
> >
> > > We introduce novel casework
> >
> > I think the technique of using stratifying spectral information has been done before, the application to this problem, I agree, is new, but so is the problem itself! Also F.2 seems pretty straightforward. TBF, I am not docking the authors or this work about these.
> >
> > > SVD at each round in the worst case
> >
> > Thanks for the clarification, I agree, the rounds where RECLUSTER needs to be called is sparse and so the cost of SVD doesn’t significantly add to the computational complexity.
> >
> > > distributional shifts
> >
> > Thank you for your comments on this. It will be great if the authors can include these discussions in the final print.
> >
> > I will look forward to your response, thanks!

---

> > > ### Author Response · Authors · 2025-11-30
> > >
> > > > The whole body of work on subspace approximations for streaming algorithms are designed to approximate subspaces in various kinds of streaming setting. So to say that FD is the only SoTA algorithm out there is confusing to me.
> > >
> > > Yes, we absolutely agree with the reviewer that FD is **not the only algorithm** for streaming low-rank approximation, nor is it necessarily state-of-the-art. However, we chose FD because it is simple to implement and, more importantly, it clearly illustrates the need for studying **consistency**, which is a different objective than standard streaming LRA. In fact, it can be shown theoretically that other simple algorithms, such as recomputing the top-$k$ subspace on every update, **without even restricting space**, would incur **linear recourse**. While our primary contribution is theoretical, we additionally perform extensive experiments across multiple values of $k$ and $\varepsilon$, and on several datasets (Landmark, RICE, SKIN, synthetic), which serve as a proof-of-concept that our algorithm is practical and achieves low recourse in realistic settings.
> > >
> > > > I think the technique of using stratifying spectral information has been done before, the application to this problem, I agree, is new, but so is the problem itself! Also F.2 seems pretty straightforward. TBF, I am not docking the authors or this work about these.
> > >
> > > Yes, we agree with the reviewer's comment that stratifying spectral information has almost certainly been done before in various contexts.  We also see that the reviewer has noted that while the *concept* of analyzing the singular value spectrum is a broadly used tool, its specific application here coupled with recourse minimization and the need to address exponential cost decay is new and necessary for achieving our bounds, as the problem itself requires simultaneously maintaining both accuracy (spectral properties) and stability (recourse), which fundamentally differs from standard oblivious sketching or streaming low-rank approximation (LRA).
> > >
> > > The statement of **Lemma F.2**, which provides a lower bound on the $(k+1)$-th singular value $\sigma_{k+1}^2(A^{(t)})$ for our working matrix $A^{(t)}$, may appear straightforward, but its necessity and context are tied directly to the **bit complexity** of the streaming environment, a property somewhat unique to this area. For general real matrices, $\sigma_{k+1}^2$ can be arbitrarily small, resulting in an arbitrary number of epochs where the optimal low-rank approximation cost doubles, thereby crashing our intended $\tilde{O}(k^{3/2})$ recourse. We overcome this by leveraging the fact that our matrix $A^{(t)}$ consists of **weighted integer vectors** with polynomially bounded entries, derived from the upstream stream reduction.
> > >
> > > Moreover, the core technical work lies not just in proving Lemma F.2, but in its **integration** into the overall analysis using a **level set argument** to achieve the $k^{3/2}$ recourse across all possibilities. The argument partitions the analysis based on the rank $r$ of $A^{(t)}$. In the **large-rank case** ($r \ge 2k$), the exponent in Lemma F.2 is small, providing a $\sigma_{k+1}^2$ lower bound that is only polynomially small, which guarantees that the number of epochs in which the cost doubles is limited to $O(\log n)$, which gives roughly $O(k\log n)$ recourse from the expensive full recomputations of the top-$k$ space. Conversely, in the **small-rank case** ($k < r < 2k$), the rank is inherently close to $k$. In this scenario, we switch strategies, proving that we can simply maintain the **entire row span** of $A^{(t)}$. Since the dimension is small ($r < 2k$), the recourse paid is for maintaining an $r$-dimensional space, which is still manageable and ensures the total recourse remains within the $\tilde{O}(k^{3/2})$ bound. This two-pronged strategy, which is enabled by the structural bound of Lemma F.2, is essential for upper bounding the recourse for matrices with large condition number. Finally, we remark that this is all still only casework for the full analysis/algorithm which further adapts its strategy based on the values of the bottom $\sqrt{k}$ eigenvalues among the top $k$ eigenvalues. Thus, we believe there is a number of technical intricacies and novelties our work introduces in order to achieve the desired results.
> > >
> > > > Thank you for your comments on this. It will be great if the authors can include these discussions in the final print.
> > >
> > > Thanks for the suggestion, we have added a discussion in Section 1.1 in the revised version, remarking that our current techniques can handle insertion-deletion updates due to Lemma 2.1. Thus we can handle explicit distribution shifts. We also note that our techniques currently do not handle implicit distribution shifts, such as in the sliding window model, and that this would be an interesting direction for future work.

---

### Author Response · Authors · 2025-11-21

We thank the reviewers for their careful and thoughtful comments. We also appreciate the positive feedback provided on our paper, including:
- The problem setting is interesting, i.e., studying of the subspace corresponding to streaming updates and understand how subspace can differ for different algorithms is an interesting idea. (Reviewer ex4U)
- Given the authors present a lower bound to the problem, it helps us ground the theoretical upper bounds presented in this work. (Reviewer ex4U)
- I really appreciate the simple algorithms which helps maintain the approximation at time $t$. (Reviewer ex4U)
- Good empirical evaluations help us understand how the algorithms presented here work in practice. (Reviewer ex4U)
- The paper introduces a novel model for studying low rank approximation of consistency. (Reviewer odAo)
- Consistent and low recourse algorithms have been studied for other problems in data science thus making low rank approximation a natural problem to study from a theoretical perspective. (Reviewer odAo)
- Moreover the authors show good upper and lower bounds for low rank approximation in this model. (Reviewer odAo)
- Standard streaming subspace approximation algorithms like FrequentDirections and Ridge Leverage Score Sampling can have very large recourse as shown theoretically, and empirically on real data. That means they can bounce between solutions. (Reviewer vaVm)
- The algorithm is subtle yet simple...with focus on ensuring a stable answer. (Reviewer vaVm)
- I think the recourse setting is natural and useful. It is a nice way to quantify stability of the sketch. (Reviewer vaVm)
- A strength is that feels like a complete paper on this topic. It has a variety of upper bounds for additive and relative error, and shows lower bounds on recourse that asymptotically match the upper bounds. There are basic experiments that show that the algorithm is not just theoretical, but works in practice -- whereas baselines like FrequentDirections does not. (Reviewer vaVm)

We have uploaded a revised version of the manuscript that includes additional discussion on the practical motivations of consistent low-rank approximation, particularly in applications such as biometrics, image processing, text mining, and large-scale data curation, where stable feature representations reduce retraining costs and improve model reliability, with the changes marked in blue. We hope these updates further highlight the compelling reasons to study this problem.

We provide individual responses to specific comments in more detail below.

---

### Meta-Review · Area_Chair_ze3f · 2026-01-07

**Summary:**

The submission studies row-arrival streaming low-rank approximation where, at every time t, the algorithm must output a rank-𝑘
subspace that approximates the optimal rank-𝑘 approximation of the prefix matrix , while also keeping the sequence of subspaces stable over time. Algorithmically, a key technique is casework on the spectral mass of the bottom k singular directions (within the top-k space) to decide between small “local” replacements versus occasional “recluster” events (SVD refresh), aiming to control how much the subspace changes per update.

**Reviewer Concerns:**

Reviewer vaVm (Rating: 8, Confidence: 5)

No weaknesses/questions

Reviewer odAo

Wants more concrete practical motivations/applications
 — Addressed: authors add expanded applications discussion (biometrics, image processing, text mining, data curation, etc.)

Reviewer ex4U
(1) Only FrequentDirections compared empirically... surprising given rich LRA literature
 — Partially addressed: authors argue their objective is consistency/recourse and FD is representative; they do not add many additional baselines but justify why FD is the relevant standard method for demonstrating “linear recourse” failure
.
(2) Theory seems “derivative”; wants clearer novelty

 — Addressed: authors give concrete technical novelties

(3) Concern Distributional shifts

 — Partially addressed: authors add discussion that insertion-deletion updates are handled, but sliding-window/implicit shifts are not, and left as future work

**Reviewer Scores:**

Reviewer ex4U is likley to revist their score given the rebuttal.

---

### Decision · Program_Chairs · 2026-01-26

Accept (Poster)